

# CFD-based curved tip shape design for wind turbine blades

Mads H. Aa. Madsen[1], Frederik Zahle[1], Sergio G. Horcas[1], Thanasis K. Barlas[2], and Niels N. Sørensen[1]

[1]Aero- and Fluid Dynamics (AFD) section, DTU Wind Energy, Lyngby Campus, Nils Koppels Allé, building 403, 2800 Lyngby, Denmark
[2]Airfoil and Rotor Design (ARD) section, DTU Wind Energy, Risø Campus, Frederiksborgvej 399, 4000 Roskilde, Denmark
**Correspondence:** Mads H. Aa. Madsen (mham@dtu.dk)

**Abstract.**

This work presents a high-fidelity shape optimization framework based on computational fluid dynamics (CFD). The presented work is the first comprehensive curved tip shape study of a wind turbine rotor to date using a direct CFD-based approach. Preceeding the study is a thorough literature survey particularly focused on wind turbine blade tips in order to place the present work in its context. Then follows a comprehensive analysis to quantify mesh dependency and to present needed mesh modifications ensuring a deep convergence of the flow field at each design iteration. The presented modifications allow the framework to produce up to 6 digit accurate finite difference gradients which are verified using the machine accurate Complex-Step method. The accurate gradients result in a tightly converged design optimization problem where the studied problem is to maximize power using 12 design variables while satisfying constraints on geometry as well as on the bending moment at 90% blade length. The optimized shape has about 1% r/R blade extension, 2% r/R flapwise displacement, and slightly below 2% r/R edgewise displacement resulting in a 1.12% increase in power. Importantly, the inboard part of the tip is de-loaded using twist and chord design variables as the blade is extended ensuring that the baseline steady-state loads are not exceeded. For both analysis and optimization an industrial scale mesh resolution of above $14 \cdot 10^6$ cells is used which underlines the maturity of the framework.

## 1 Introduction

The wind energy industry has for decades focused on minimizing the levelized cost of energy (LCoE) (Ning et al., 2014; Kalken and Ceyhan, 2017; Matheswaran et al., 2019). Innovative design has helped to keep the mass increase low as rotor size has increased to generate more annual energy production (AEP) which in turn lowers the LCoE. While manufacturing new wind turbines with diameters now exceeding 200 m helps the industry meet the present day energy demand one should also look to already installed wind turbines which may hold promise for an increase in AEP as well. A first approach could be to completely refurbish these older wind turbines with new rotors although the associated costs are considerable. A less invasive operation would be to only modify the very tip of the blades with a sleeve-like solution. Indeed, using this approach design engineers have successfully met the challenge of avoiding to compromise the integrity of the original rotor and at the same time found an increase in power of up to several percent and thus, a further investigation is warranted (Zahle et al., 2018; Matheswaran et al., 2019; Barlas et al., 2021).





In light of the above, this study investigates the promise of re-designing existing wind turbine blades by optimizing the shape of the blade tip. To maintain structural feasibility the performance gain will be brought about without exceeding the load envelope. The present work should be placed in the context of the SmartTip project which in several efforts (e.g., Zahle et al. (2018); Li et al. (2018); Barlas et al. (2021)) studied innovative blade tip modelling and design, in particular how blade
tips could yield a load neutral performance gain. Zahle et al. (2018) find a 2.6% increase in power using 12 design variables with an aerodynamic surrogate model whereas Barlas et al. (2021) report up to 6% increase in AEP using 11 design variables for an aeroelastic surrogate approach. Given that the above-mentioned works vary in included disciplines and model fidelities one should take care when comparing the final results. However, since the present study uses the exact same overall design problem which was addressed in the surrogate-based design study by Zahle et al. a comparison across flow model fidelities
is given later on. The present study should in this context be seen as a contribution considering aerodynamics only but with a direct high-fidelity modelling approach using a CFD solver. Thus, the aim for the present study is to test the developed CFD-based design framework using the finite difference method in an industrial scale setting for the first time to quantify how viable this approach is and if its limitations are outweighed by the ease of implementation associated with the method. Reasons for choosing a CFD-based approach with a gradient-based optimization algorithm are given below after which the content of the
remaining paper is outlined.

Lower-fidelity methods based on the blade element momentum (BEM) theory rely on engineering assumptions and can as a result handle time-dependent simulations efficiently which explains why they are relied on heavily in the wind energy community. High-fidelity CFD-based approaches, on the other hand, allow for investigations that do not depend on underlying engineering assumptions by solving the full Reynolds-averaged Navier–Stokes (RANS) equations on a geometrically resolved
rotor configuration. Directly resolving the geometrical features at the blade tip ensure a correct modelling of the highly complex 3-D flow phenomena at the blade tip. However, also disadvantages such as an increase in computation time as well as in implementation effort is to be expected. Furthermore, as high-fidelity models typically are used in steady-state it is currently difficult to arrive at realistic design driving load cases with this approach. For these reasons, one should favour a complementary use of the two approaches and only use high-fidelity methods if needed. Given, that the area in question in the present study,
i.e., the blade tip, is indeed an area of difficulty for lower-fidelity approaches the use of a CFD-based model is warranted despite the increase in computation time.

To accurately describe the blade tip it was in the present study necessary to use 12 design variables. This is a considerable amount of design variables compared to many of the other tip studies mentioned below which typically use less than 5. Due to the increased size of the parameter space a gradient-based optimization algorithm was preferred. Moreover, a proper step size
was chosen through a gradient verification study using the Complex-Step method (Lyness, 1967; Lyness and Moler, 1967) as a machine accurate reference gradient. The accurate gradients led to a tight optimization problem convergence.

While the presented CFD-based approach using the finite difference method is well-functioning, it is not feasible to undertake a full simultaneous design of airfoils and blade planform. For this, the associated cost of computing the gradient is too high since 3-D shape optimizations of full rotor configurations involve hundreds of design variables (Nielsen and Diskin, 2012; Madsen
et al., 2019). To carry out a design optimization study of a full rotor configuration using finite-difference based gradients, one





could apply a more conventional approach where a sequential procedure in two steps should work (Barrett and Ning, 2018): First, the airfoils could be optimized with a method of choice (e.g., a panel method). Then, the planform could be optimized using the present CFD-based methodology. In practice, however, one would likely need to iterate between the two steps to arrive at a final design. See Barrett and Ning (2018) for further details and alternative approaches.

The remaining paper is structured as follows: A literature review of shape optimizations focusing on wind turbine blade tips is given in Sec. 2 whereafter the methodology used in the present study is presented in Sec. 3. Then follows a comprehensive analysis of the baseline rotor (Sec. 4) followed by a presentation of the design optimization problem (Sec. 5) before all optimization results are presented in Section 6. Finally, a conclusion is given (Sec. 7) where overall findings are summarized.

## 2   Literature review

This literature review focuses exclusively on shape optimization of the blade tip from purely aerodynamic works. A literature review on general high-fidelity shape optimization of full rotor configurations within wind energy research is offered elsewhere (Madsen et al., 2019, Sec. 2) where an updated overview table can be found in a more recent work (Madsen, 2020, Tab. 3.1). With respect to literature surveys focused on tips and winglets there are already a few present in the wind energy community (e.g., Gertz et al. (2012)). However, some are several years old and none are as comprehensive as the below-given literature

survey. While some of the works mentioned below are indeed experimental, the focus is on numerical studies given that also the present work is purely numerical. A literature survey including further experimental studies is presented by Mühle et al. (2020).

In order to structure the covered works the survey is split into the sections;

    - Early works on wind turbine blade tips (Sec. 2.1),

- Parametric studies (Sec. 2.2), and

    - Design optimization studies (Sec. 2.3),

before overall conclusions are presented in Sec. 2.4.

For an up front overview of all central works for the present study across the above mentioned subsections one can consult Tab. 1. From this table it is possible to obtain rough estimates of, e.g., what a reasonable performance increase for curved tip

shapes is. It is also evident from Tab. 1 that only few works on industrial scale wind turbines exist where in particular actual optimization works seem to be few in number.

Care must be taken when comparing all the works listed in Tab. 1. Many works (Hansen and Mühle, 2018; Khalafallah et al., 2019; Khaled et al., 2019; Mourad et al., 2020; Papadopoulos et al., 2020; Aju et al., 2020) are focusing on smaller turbines whereas only three other works (Kalken and Ceyhan, 2017; Matheswaran et al., 2019; Zahle et al., 2018) focus on

larger MW turbines as is the case in the present study. Furthermore, as will be evident in Sections 2.2-2.3 there are few available actual optimizations whereas the majority of works are parametric studies and thus better performing tips could very likely be found in the vicinity of the selected parameter settings for these rotor configurations. Still, Tab. 1 does provide the reader with an overall survey of the related available literature and should help temper the expectations with respect to the



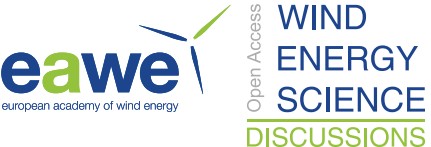

**Table 1.** Overview of the tip and winglet studies most relevant to the present work. Gray rows signify a reported load neutral final design. In case of a sequence of studies by the same author(s) a representative study was chosen. For works including both a simple straight blade extensions and an actually novel tip device including a flapwise displacement it is the latter that has been included below.

| Reference | Dir.⋆ | Fidelity | Turbine‡ | Mesh size † | Optimi-zation | Design variables | Improvement | Objective |
|---|---|---|---|---|---|---|---|---|
| Johansen and Sørensen (2006) | ++ | CFD | - | $1.4 \cdot 10^7$ | - | 2 | 1.4% | Power |
| Johansen and Sørensen (2007) | SS | CFD | - | $1.8 \cdot 10^7$ | - | 4 | 2.8% | Power |
| Ferrer and Munduate (2007) § | ÷ | CFD | NREL VI | $1.2 \cdot 10^7$ | - | 2 | 20.1% | Torque/Thrust |
| Elfarra et al. (2014) § ¶ | SS | surrogate ¶¶ | NREL VI | $(7.0 \cdot 10^5)$ | ✓ | 2 | 9.0% | AEP |
| Aravindkumar (2014) | PS | field test | SHWT | - | - | 1 | 2.0% | Power |
| Tobin et al. (2015) | SS | wind tunnel | SHWT | - | - | 1 | 8.2% | Power |
| Ariffudin et al. (2016) | ++ | CFD | SHWT | $6.8 \cdot 10^6$ | - | 1 | 3.2% | Power |
| Zhu et al. (2017)¶ | ++ | CFD | SHWT | $1.8 \cdot 10^7$ | - | 1 | 4.0% | Power |
| Kalken and Ceyhan (2017) | PS | vortex code | N80 2.5 MW | - | - | 3 | 2-9% | Power |
| Hansen and Mühle (2018) § | SS | surrogate §§ | SHWT | $(1.3 \cdot 10^7)$ | ✓ | 6 | 7.8% | Power |
| Zahle et al. (2018) | PS | surrogate ⋆⋆ | IEA 10 MW | $(6.0 \cdot 10^6)$ | ✓ | 12 | 2.6% | Power |
| Reddy et al. (2019) § | ++ | surrogate ⋆⋆ | Vestas27 | $(3.6 \cdot 10^7)$ | ✓ | 5 | 4.5% | Power |
| Matheswaran et al. (2019) | SS | vortex code | NREL 5 MW | - | - | 4 | 2.5% | Power |
| Khalafallah et al. (2019) § | ++ | CFD | SHWT | $5.4 \cdot 10^6$ | - | 4 | 4.4% | Power |
| Khaled et al. (2019) § | PS | surrogate ¶¶ | SHWT | $(1.0 \cdot 10^7)$ | ✓ | 2 | 6.3% | Power |
| Farhan et al. (2019) | SS | CFD | NREL VI | $1.7 \cdot 10^7$ | - | 3 | 9.8% | Power |
| Mourad et al. (2020) | ++ | CFD | SHWT | $3.8 \cdot 10^6$ | - | 2 | 2-6% | Power |
| Sy et al. (2020) § | SS | CFD | NREL VI | $2.8 \cdot 10^6$ | - | - | 1.2% | AEP |
| Papadopoulos et al. (2020) § | ++ | CFD | SHWT | $1.4 \cdot 10^7$ | - | 2 | 10.9% | Power |
| Madsen et al. (present work) | PS | CFD | IEA 10 MW | $1.4 \cdot 10^7$ | ✓ | 12 | 1.1% | Power |

⋆ Whether the tip is directed towards the suction side (SS) or the pressure side (PS). When both directions were investigated the symbol is ++ and for tips in the rotor plane without any flapwise displacement the symbol, ÷, is used.

§ Due to their periodic boundaries the reported mesh size has been multiplied with number of blades.

¶ Only the amount of mesh vertices are listed not mesh cells.

† Number of cells in largest mesh used for optimization. Parenthesis signify that the mesh pertains to an underlying training material.

‡ Small horizontal axis wind turbines (SHWT) are smaller wind turbines of various configuration up to about 1 m in diameter (Ariffudin et al. (2016) report a 4 m diameter). Winglets have been reported to have effect up to 50 m inboard (Zahle et al., 2018) meaning that one should to care when combining SHWT-works with the remaining works.

⋆⋆ Response surface.

§§ Kriging model.

¶¶ Artificial neural network (ANN).



possible performance increase when optimizing the tip. Importantly, there are three works (gray rows) that manage to provide
a performance enhancing tip design that does not violate the initial load envelope. These studies can therefore not be compared
directly to the studies without load constraints. Finally, it should be noted that a clear progression in level of model fidelity can
be seen over time. Thus, many of the later works rely exclusively on CFD allowing researchers to analyze the finer details of
the 3-D complex flow phenomena present at the blade tip regions.

## 2.1 Early works on wind turbine blade tips

The concept of a winglet can be dated more than a century back to the English engineer Frederick W. Lanchester's 1897
patent application for fixed-wing aircrafts (Lanchester, 1897). However, more recent history picks up in the 1970s when R. T.
Whitcomb (Whitcomb, 1976) further refined the idea[1]. It has been known since the pioneering days of Whitcomb that even
a small winglet can limit the spanwise velocity component and reduce downwash by displacing the tip vortex which ideally
diffuses or is smeared out (Whitcomb, 1976). Furthermore, with respect to the winglet orientation Whitcomb says that lower
(i.e., pressure side) winglets should be as effective as upper (i.e., suction side) winglets (Whitcomb, 1976, p. 8). In aerospace,
the former are typically the smaller ones due to concerns of ground clearance. If anything, this is quite the opposite for wind
turbines where it is the winglet on the suction side, i.e., an 'upper' winglet in aerospace terminology which for wind turbines
can cause a tower strike and therefore should be kept small. Another considerable difference is that the system of interest in
wind energy is rotating which adds some complexity to the induced velocity seen by the blade from the tip vortex. There are
also other differences to study when transitioning from one sister science (aerospace) to another (wind energy). For a concise
recap of essential works on winglets for fixed-wings one can consult the introduction in Gaunaa et al. (2011).

About a decade after Whitcomb's 1976 study, Lissaman and Gyatt (1985) present the perhaps first comprehensive study for
wing tip devices dedicated to wind turbines. As seen from their reference list a considerable amount of research on these tip
devices had already been carried out in the aerospace community at this point. However, as Lissaman and Gyatt point out, no
studies focus directly on wind turbines. Using both field testing and numerical analysis codes (vortex methods) they analyze
three tip shapes. However, they find that none of the shapes resulted in actual performance improvement. Interestingly, they
also focus on noise and report that both the standard winglet as well as the split winglet produce significantly more noise
compared to the baseline configuration which they attribute to added turbulent flow. For future improvements, Lissaman and
Gyatt see a need for increased model fidelity in the used computer code and, as a result, they believe the most cost-effective
approach is further field- or wind tunnel tests. Furthermore, means of visualizing the flow seems of interest. Not surprisingly,
such techniques have more recently been heavily used, such as, smoke (Mühle et al., 2020, Fig. 6), particle image velocimetry
(Aju et al., 2020, Fig. 9), and oil (Andersen et al., 2001, Fig. 7).

More than a decade after the earliest cited wind energy work (Lissaman and Gyatt, 1985) one will find the first significant
increase in numerical model fidelity where Madsen and Fuglsang (1997) present actual CFD-based results of novel tip designs.
This is to the best of the authors' knowledge the earliest 3-D CFD-based study. Madsen and Fuglsang motivate their use of a
high-fidelity CFD model by stating that the BEM-based approaches are more uncertain in the tip region due to complex flow

---

[1]https://appel.nasa.gov/2014/07/22/this-month-in-nasa-history-winglets-helped-save-an-industry/, accessed Sep 30, 2021





phenomena. Using the presented methodology they are able to present a design with a nonseparating tip vortex thus lowering the tip noise. Thanks to the higher model fidelity, they can use streamlines to visualize the finer details of the flow. This allows them to align the twist angles of the winglet to match the streamlines thereby avoiding separation (Madsen and Fuglsang, 1997, Fig. 6-11). Impressively, the final design was also tested in an experimental setup (Andersen et al., 2001) a few years later, thus lending further credibility to the results.

Imamura et al. (1998) use a vortex lattice method with free wake modeling to compare the performance of rotors with and without winglets. Five different downstream winglets were tested as well as a baseline blade and a pure tip extension. The numerical results (Imamura et al., 1998, Fig. 10) show that all winglets indeed have a beneficial effect on the power coefficient, $C_p$, where no quantification is given. However, the pure blade extension is not far removed from the baseline performance. It should be noted that increases in bending moment are also reported (Imamura et al., 1998, Fig. 11). Again, there is no quantification. This effect is worst for straight extensions and the bending moment increase is seen to be somewhat mitigated as the winglet increases its bending towards the downstream direction.

Already at this point, many of the key topics evident throughout the literature have been presented: It is not straight forward to gain a performance increase unless leveraging a meticulous design. Even in the cases where this is achieved, there are many other characteristics such as noise production which are important to consider. Perhaps for this very reason, many works start to focus on a select few parameters in order to narrow down the related effect for each parameter change. This trend is particularly evident in the next section.

## 2.2 Parametric studies

In the beginning of the 2000s, more works of medium- and high-fidelity models start to emerge. The vast majority of the works covered in this literature review are so-called parametric studies which also are known as design of experiments. For these works, the parameters are methodically varied in order to explore the underlying design space. Several works use the terms 'optimizer' and 'optimization' rather loosely and thus, some works below (Farhan et al. (2019); Papadopoulos et al. (2020), etc.) will describe their work as optimizations. However, if the design parameter variations are chosen beforehand and there is no individual optimizer component that traverses the design space they will in the present literature survey be categorized as parametric studies.

Johansen and Sørensen (2006) study five wind turbine configurations with different camber and twist distributions in a parameter study using CFD. All five configurations had the exact same chord distribution. Both upstream and downstream winglets were analyzed where the latter type is found to outperform the former. They use cant angle and sweep angle to describe their winglet geometries, where cant angle refers to the angle of incidence the flapwise displaced part (i.e., the out-of-plane part) of the tip makes with a straight blade reference axis (i.e., a $0°$ cant angle is a straight blade and $90°$ is a true winglet). Similarly, the sweep is the incident angle for the part of the tip that lies in the rotor plane. In this study cant angle ($90°$) and sweep ($0°$) is not altered but merely used to describe the designed winglets. The various tested twist and camber settings result in power increases of $0.6 - 1.4\%$ whereas trust increases $1.0 - 1.6\%$ for winglets of about 1.5% height.





Not long after, Johansen and Sørensen (2007) again carry out a CFD-based parameter study this time showing that the tested winglets bring about a $1.0 - 2.8\%$ increase in power at the cost of $1.2 - 3.6\%$ increase in thrust. The varied winglet parameters are height, curvature radius, sweep, and twist having the following definitions: Height refers to the distance that the winglet protrudes in the out-of-plane region (i.e., the distance to the rotor plane). Curvature radius is a related measure given in percentage of the winglet height which describes how smooth the transition from straight blade to winglet occurs (0% means

a $90°$ kink and 100% means a very smoothly transitioning blade that first at the very tip of the winglet reaches the maximum projected blade length). In light of their previous results (Johansen and Sørensen, 2006) they only test downstream winglets. They test a total of 10 different rotor configurations. In relation to the present study it is relevant to mention that they find no effect on power for sweep and that only limited effect from changing the twist can be observed.

Ferrer and Munduate (2007) also use CFD and analyse three tip configurations for the NREL Phase VI wind turbine rotor.

The three configurations are a square tip, a highly tapered tip where the tip ends at the pitch axis, and a highly tapered tip where the trailing edges were aligned resulting in a swept-back tip. Here, the pitch axis should be taken as the quarter-chord location. They find that both tapered tips outperforms the rectangular tip which has a large tip vortex. Of the tapered tips it is the configuration with its tip on the pitch axis that has the best torque to thrust ratio. Furthermore, they specifically point to a complementary use of CFD in lower-fidelity design approaches.

Gaunaa and Johansen (2007) show that the increase in $C_p$ resulting from a winglet is owed to a reduction in tip effects (i.e., tip loss) and not as previously thought due to a downstream shift of the wake vorticity. However, in the same work a comparison between the developed free wake lifting line model and a CFD reference was not entirely successful leading to a follow-up study (Gaunaa and Johansen, 2008) where said comparison was improved. In both works they advocate for downstream winglets which they find to be more efficient than their upstream counterparts.

A few years later, Gaunaa et al. (2011) use computationally lighter models based on lifting line theory to analyze blades with winglets. The tested models are a free wake model and a much faster prescribed wake model. The study relates highly to previous work by the authors (Gaunaa and Johansen, 2007) where the free wake model was used. Moreover, the validation of the developed prescribed wake model is carried out against CFD-based results from Johansen and Sørensen (2006). By comparing results from both the free wake model and the prescribed wake model to CFD results they conclude that these

faster model types successfully can predict the effect of adding winglets to wind turbine rotors (errors: $4 - 16\%$ and $17 - 28\%$, respectively (Gaunaa et al., 2011, tab. 2)). Considering how well the wake models approximate the results from the CFD solver it would be very interesting to see these codes applied in an optimization context in future works.

Using high-fidelity models Sørensen et al. (2011) sought a deeper understanding of the underlying flow phenomena related to several tip shapes. This is, e.g., a necessity when a thorough understanding of the generated noise level is needed. Furthermore,

they used access to both high- and lower-fidelity models to carry out a detailed comparison across fidelities. This allowed Sørensen et al. (2011) to implement an improved tip correction for lifting line models.

Aravindkumar (2014) use experimental field tests to compare with a CFD model in order to investigate performance increase and noise reductions. They find that adding an upstream winglet increases the generator power with 2.0% whereas the noise is reduced with 25 %.



Tobin et al. (2015) use wind tunnel experiments to compare one rotor configuration without a winglet with a rotor configuration including a winglet and observe a $C_p$ increase of 8.2%. A related thrust (i.e., $C_t$) increase of 15.0% was observed. Having studied the literature they see a need for further insight in wake performance enhancement resulting from winglet design. They find that winglets do not significantly alter tip vortex strengths.

Ariffudin et al. (2016) use a CFD model to investigate four different configurations; a straight extension, a swept extension that lies in the rotor plane, and configurations with winglets directed either upstream or downstream. Between the two first configurations it is the swept configuration that outperforms the straight configuration with a 9.1% and a 7.3% $C_p$ increase, respectively. As for the winglets it is the downstream configuration that results in highest performance increase with 3.2% compared to just 1.8% increase for the upstream winglet. No quantification of similar changes in the thrust coefficient, $C_t$, is offered.

Zhu et al. (2017) carry out a study aiming at determining the best direction for a winglet. Upstream-, downstream-, and even split winglets are tested. The split winglet pointing both upstream and downstream is found to be the best performing device and a $C_p$ increase of up to 4.0% is observed. Comparing just the pressure side and suction side winglets the former outperforms the latter since the pressure side winglet results in up to 3.8% increase whereas the suction side winglet only reach a 3.4% increase (Zhu et al., 2017, Tab. 5). They also conclude that the angle of the winglet should match the incoming flow angle as much as possible (something focused on already in the very early works referenced above (Madsen and Fuglsang, 1997)). Finally, they mention that an actual optimization would further increase the performance.

Kalken and Ceyhan (2017) study three different tip concepts (turbulators, winglet, and conventional tip) and submit the final designs to experimental testing on a 2.5 MW wind turbine for a final validation. The early design phase leverages BEM and lifting line models combined with free wake models whereas CFD is used as an analysis tool on the finalized tip design. Kalken and Ceyhan (2017) report based on measurements that more than 4% power increase is owed to a simple blade extension whereas the benefit of using different blade tip shapes result in 2-9% power increase. The CFD analysis shows that a conventional extension results in a higher power to thrust ratio gain than the studied winglet. However, they point to other beneficial side effects for choosing a winglet (noise reduction, height restriction, etc.).

The work by Matheswaran et al. (2019) is particularly interesting as it is one of the rare studies with a reported load neutral tip design. This recent work has even resulted in a patent[2] stressing the industrial relevance of novel tip designs that do not compromise the load envelope for the baseline wind turbine. Matheswaran et al. (2019) motivate their study by stressing the point that most tip designs for wind turbines do not focus on e.g., the added flapwise bending moment. As a result, the initially minimally invasive operation of retrofitting the very tip of the blade may become an intractable proposition altogether. By balancing the centrifugal force with the aerodynamic forces generated by the winglet itself they present a lightweight winglet which focuses on minimizing the bending moments. The presented design methodology is based on the vortex lattice method which, unlike traditional BEM approaches, manage to model the flow at the very tip of the blade. The model has a prescribed wake although Matheswaran et al. (2019) do point out that a 'free wake' approach would be ideal, albeit also computationally

---

[2]https://patentscope.wipo.int/search/en/detail.jsf?docId=US242151286&recNum=7&docAn=16186876&queryString=(IC/F03D)%20&maxRec=92796, accessed Sep 30, 2021. USPTO. Application number 16186876





more demanding. After having validated their vortex code against both experimental and numerical results they introduce an LCoE cost model and carry out a parametric study using four parameters (height, taper ratio, as well as twist- and cant angles).

Only suction side winglet designs are investigated. Having studied various configurations they decide on a design with the best compromise between a respectable increase in $C_p$ (2.5%) while the force ratio between aerodynamic loads and centrifugal forces is kept manageable (1.4-1.8). They carry out a structural analysis to prove the designs can withstand the required loads. For future work they point to the need for higher model fidelity where they specifically mention using a CFD model.

Khalafallah et al. (2019) present a parametric study of winglets' possible effect on wind turbine power production. Both

swept and straight blades are tested using a CFD model with up to 1.8 million cell meshes with periodic boundary conditions modeling only one of three blades for the rotor. Upstream- and downstream winglet configurations with various cant angle and twist angle are tested and the best result is achieved for the upstream configuration with a 4.4% power coefficient increase. Using results from a previous study they select a few well-performing swept baseline blades and investigate the effect of adding winglets on the resulting power coefficient. 18 straight blade configurations and 33 swept blade configurations were

simulated. In general, they find that downstream winglets outperform upstream winglets for straight blades (Khalafallah et al., 2019, Tab. 3). However, for swept blades (Khalafallah et al., 2019, Tab. 4-5) the conclusions are less unimodal. Indeed, the best swept winglet design is an upstream directed winglet yielding a 4.4% increase in $C_p$. Overall, they conclude winglets can indeed be used to enhance rotor performance and that they may as well lead to a reduced thrust coefficient. It should be noted that a reduction in thrust coefficient is only observed in the comparison to a swept baseline blade (Khalafallah et al., 2019,

Fig. 13) whereas as it is quite clear that the coefficient of thrust for the winglet coefficients in general increase compared to the straight blade baseline.

Farhan et al. (2019) investigate winglet planform and airfoils' importance for winglet design using CFD. They vary height, cant angle, and planform using two different airfoils and find that the best performance increase is owed to about a distance equivalent to 3% of the blade length, which from now on will be written as 3% d/R for brevity. The configuration had a

rectangular winglet with a $45°$ cant angle. To choose the best turbulence model they start out by comparing two RANS models, i.e., the Spalart-Allmaras and the Shear Stress Transport (SST) model by Menter (1992), and find that the latter outperforms the former in emerging stall regimes. Finally, they test two airfoils and vary cant angle and winglet length (24 combinations) and report a -13.5 % to +9.8 % change in power depending on the wind speed and configuration (Farhan et al., 2019, Tab .3). The tested winglets were pointing in the downstream direction. Having chosen winglet height and cant angle, they then fix the

airfoil shape to S809 and study planform effects (rectangular vs. elliptical winglets). They find that both winglet length and cant angle are amongst the most important design variables for improving performance.

Mourad et al. (2020) use an initial literature survey covering 10 references to conclude that i) there is no agreement on optimum winglet configuration, ii) height is the most effective winglet parameter, and iii) that upstream directed winglet should be preferred over downstream configurations. Since they found no study covering toe angle, i.e., the angle of attack

between tangential velocity component and winglet profile, they carry out a parameter study using height and toe angle as their two parameters. The analysis showed that of the winglet heights ranging from 0.8-8.0 % d/R it was the smaller winglet height of 0.8 % d/R that gave the largest power increase. Furthermore, Mourad et al. (2020) report that a downstream directed



winglet is not useful for wind turbine rotors. The poor performance of the winglet with 8% height is not easily aligned with previously reported results by Gertz and Johnson (2011) and Gertz et al. (2012) stating a 5% increase in power for that winglet
height - albeit for different overall parametrization. Finally, it is found that from toe angles ranging from $-30°$ to $+30°$ it is the (upstream directed) $+20°$ toe angled winglet that have the most beneficial effect where a 2-6% increase in power coefficient is observed depending on the tip speed ratio. However, a related increase in thrust coefficient of 4.6-9.8 % is reported meaning that the initial load envelope is compromised.

Kulak et al. (2020) present both experimental and numerical results in a study of a small wind turbine rotor (20 cm radius)
where the aim is to raise overall power output. Wind tunnel tests of configurations with and without winglets were carried out to quantify the differences. In total, one baseline configuration, one suction side winglet (4% height), and 3 pressure side winglets (3%, 4%, and 5% height) were tested. The numerical results for the comparison were only generated on the pressure side winglet with 4% height as well as for the baseline rotor. A nice detail in this study is that they add a transitional model to increase model fidelity. In agreement with Gupta and Amano (2012) an increase in $C_p$ for particularly pressure side
winglets is observed where up to 6% increase is reported. No efficiency increase is seen for suction side winglets. Instead, a decrease in $C_p$ is observed compared to the baseline performance. Relating the experimental results to the numerical findings show a misalignment since the numerically investigated pressure side winglet performs worse than the baseline rotor. Thus, experimental results do not agree with numerical results. They conclude that the precise winglet geometrical features must be carefully defined in order to gain the desired performance increase.

Sy et al. (2020) use CFD to study a split winglet design and its ability to lower the induced drag. In total, four different designs are analyzed (baseline, straight, suction side winglet, and split winglet). The three modified meshes all had a 1.5 % r/R extension and winglets had a $45°$ angle offset. They observe a 1.23% and a 2.53% increase in power for ordinary winglet and split winglet, respectively. However, also thrust increases with 0.83% and 2.05% percent for the designs in question.

Mühle et al. (2020) investigate the promise of using winglets to enhance wake recovery using an experimental setup. These
types of studies view a wind turbine not only as an isolated system but also as part of a whole, e.g., a wind farm. Thus, improved wake recovery in one turbine will lead to performance increase in the next one. The investigated rotor is a two-bladed model scale rotor where the wing tip can be exchanged with a downstream winglet tip to compare performance characteristics. This is the same rotor which was investigated by Hansen and Mühle (2018). The wind tunnel measurements show that a winglet not only can be used to increase power production but also to provoke earlier tip vortex interaction resulting in a faster wake
recovery.

Papadopoulos et al. (2020) perform a numerical analysis of six rotor configurations of a small rotor. They set twist and toe angle to $0°$ on the basis that available literature point to a minimal effect. The winglet height was set to 5% blade radius. They find that the cant angles of $±45°$ are better than the $±90°$ angles and that sweep only has a limited effect. However, as they also document (Papadopoulos et al., 2020, Fig. 2) the corresponding projected in-plane area of the rotor is slightly increased
for $±45°$ angles which should give these configurations an advantage. A nice detail in this study is the use of a four-equation transition model allowing for finer flow modeling than assuming a fully turbulent flow. The maximum observed effect was a 11% increase in $C_p$. For the configurations that did not increase the projected area the increase in $C_p$ (3.7%) was more modest.





Unfortunately, the impact on thrust is not reported in this study. With respect to the discussion on suction side versus pressure side winglets it is worth mentioning that the above-mentioned two configurations are pressure side and suction side winglets, respectively ($+45°$ and $-90°$). They point to future studies on additional parameters in order to further the understanding on winglet parametrization.

Aju et al. (2020) use an experimental setup to investigate the promise of using downstream winglet pitching as a means to lower turbine rotation and reduce thrust coefficient. This has relevance when, e.g., wind turbines should be protected during extreme weather conditions. Given, that the winglet mass only makes up 1.8% of the blade the investigated method should provide a much faster response time compared to pitching the entire blade. Also, winglet pitching is shown to accelerate flow recovery in wake regions. Aju et al. (2020) agree with Mühle et al. (2020) in that there is great promise for winglet use in wake recovery.

Of the numerous parameter studies mentioned above many winglet salient features such as power enhancement and load mitigation have been investigated. However, also several contradicting works were found. Much can be owed to a misalignment in the studies and one efficient way to unify the efforts would be to agree on design optimization problems to solve for. Not surprisingly, the need for actual optimizations is also brought up in some of the works (e.g., Zhu et al. (2017)). In the next section these few but important works in the literature are discussed.

## 2.3 Design optimization studies

Design optimization studies often demand meticulous implementation with a great attention to detail making them few in number in the wind energy literature. Still, some can be found. In fact, the very first work including CFD (Madsen and Fuglsang, 1997) also include both singlepoint and multipoint shape optimizations. However, by far most of the optimization studies are more recent works carried out within the last five years as detailed below.

From 2011 to 2015 a series of studies emerge where CFD is used to investigate possible winglet shapes on the NREL Phase VI rotor which has a 10 m diameter (Elfarra, 2011; Elfarra et al., 2014, 2015). Given that one of the studies (Elfarra et al., 2015) does not contain an actual optimization the focus will be placed on the other two works. In both studies containing optimizations (Elfarra, 2011; Elfarra et al., 2014) a direct CFD-based approach is deemed too computationally expensive and surrogate models in the form of artificial neural networks are favoured along with a gradient-free genetic optimization algorithm. Using 24 CFD evaluations to train the artificial neural network they are able to carry out winglet optimizations using cant and twist angle as design variables and obtain about 9% increase in power. A multipoint objective is targeted which gives a more robust design. However, as seen in Tab. 1 it has a rather coarse mesh resolution. While this may be due to the fact that it is indeed an early work it is fair to contemplate whether the used meshes are of adequate resolution as training material for the surrogate. After the optimization, a comparison of the flow characteristics for the baseline winglet design and the final winglet design reveal that the new design manages to attach the flow farther outboard resulting in the reported improvement. The ensuing parameter study (Elfarra et al., 2015) focuses on power enhancement using 16 winglet configurations to study how winglet direction, sweep, cant angle and twist influence the generated power. After having validated the numerical setup against experimental data they test the 16 configurations whereafter they conclude that suction side winglets generate more power than



pressure side winglets. Depending on configuration and wind speed they observe up to 10.43 % power increase (Elfarra et al., 2015, Tab. 6). Interestingly, this particular configuration did not have a $90°$ winglet but a $45°$ winglet. The winglet was twisted $+2°$ (towards lower angle of attack). In general, this study sequence exemplifies how high-fidelity models such as CFD solvers
can be used to visually inspect the flow and analyse the underlying flow phenomena at play.

Another sequence of relevant works is that by Hansen (2017) who applies evolutionary optimization algorithms in winglet design. The two works, Hansen (2018) and Hansen and Mühle (2018), focus on wind turbine airfoil and winglet design, respectively. The resulting optimization framework from the airfoil study (Hansen, 2018) is used in the final winglet study (Hansen and Mühle, 2018) to prepare the new airfoils needed for the ensuing wind tunnel tests. In the winglet optimization
study (Hansen and Mühle, 2018), a Kriging surrogate model is trained on CFD evaluations of a model-scale wind turbine with a winglet. The optimization involve 6 design variables and a total of 100 shapes are traversed. Finally, the wind turbine performance with and without winglet is validated using a 3-D printed experimental model in the NTNU wind tunnel[3]. Power is numerically predicted to increase 7.8% and a subsequent wind tunnel experiment of the winglet reports a 8.9%-10.3% power increase depending on the inflow turbulence. The 7.8% power increase came at the cost of a related 6.3% increase in thrust.
Hansen and Mühle (2018) point out that a better shape could be obtained by including more design variables to further refine the design parameter space.

Zahle et al. (2018) carry out a surrogate-based tip study where the aim is to increase AEP without increasing the load envelope for the baseline blade considering only steady-state normal operating conditions. It is well to clarify here that they optimize the tips for improved AEP but the final results are reported as improvements in power. Using a gradient-based ap-
proach to efficiently manage the 12 design variables they achieve a 2.6 % and a 0.76% power increase for winglet-like and straight tip extensions, respectively. These numbers are not far off compared to the report made several decades before by Whitcomb (1976) stating that a winglet can result in a lift-to-drag ratio improvement more than twice that of a straight blade extension (Whitcomb, 1976, p. 13). The surrogate-based approach allows for a very efficient optimization process although an added difficulty is that the final design may prove to (slightly) violate the bending moment for the underlying CFD model. This
study has the same design optimization problem as is used in the present CFD-based design study which is further explained in Sec. 5. Zahle et al. (2018) also study the effect each of the three tip-dedicated design variables has on the mechanical power and find that the achievable mechanical power improvement should increase as curvature, flapwise displacement and edgewise displacement (i.e., sweep) of the tip is increased. Noticeably, the underlying CFD model predicts a maximum for the sweep design variable around 2% d/R (Zahle et al., 2018, Fig. 4). Interestingly, the efficient surrogate procedure allows them to ex-
plore the Pareto curve between bending moment increase and increase in mechanical power. Their mesh resolution of 5.97 million cells combined with the 300 sampling points allow for a detailed analysis of the parameter space and the surrogate exhibited below 2% error for both torque and bending moment. Their overall workflow include tools for surface and volume mesh generation besides the exact same flow solver, EllipSys3D (Michelsen et al., 1992, 1994; Sørensen, 1995), as is used in the present study. All investigated tip shapes pointed upstream towards the pressure side of the blade to avoid a tower-strike.
Overall, Zahle et al. (2018) find that the CFD evaluations agree fairly well with the surrogate-based model. Furthermore, they

---

[3]https://www.ntnu.edu/ept/laboratories/aerodynamic, accessed Sep 30, 2021





conclude that a winglet-like extension should be favoured over a straight extension although the advantage is attenuated for higher wind speeds (Zahle et al., 2018, Fig. 6). Consequently, the predicted 2.6% increase in mechanical power is valid for the lower wind speeds (6 and 8 m/s). Finally, they explain how the tip shape in itself does not drastically increase power production but that its role instead is to diffuse and move the tip vortex further away in order to lower the induced drag.

Reddy et al. (2019) use high-fidelity CFD evaluations to train a surrogate response surface and carry out a shape optimization study resulting in a $C_p$ increase of 4.5 % while only introducing a minor thrust force penalty. Before undertaking the actual optimization they validate their numerical setup by comparing to experimental results. The defined objective is a compound function including coefficient of power, $C_p$, coefficient of thrust, $C_t$, and the twisting moment around the blade axis. They use five design variables: span, twist-, dihedral- and sweep angle as well as taper ratio. As stated by the authors, it seems to

be the first published multi-objective optimization study for wind turbine winglets. Indeed, as seen in Tab. 1 only very few actual optimizations can be found in the related wind energy literature. Reddy et al. (2019) state that they use the much faster surrogate methods since an actual CFD evaluation of each configuration is too expensive. To train their surrogate 50 initial blade designs are used. In comparison, Zahle et al. (2018) use 300 CFD evaluations to train their surrogate. However, Zahle et al. (2018) also use 12 design variables and as a result would have to use more evaluations to train their surrogate since the

underlying parameter space increases in size from 5 to 12 design variables. Based on their study, Reddy et al. conclude that winglets can increase power production and that their optimization framework is capable of handling the multidimensional design.

     Khaled et al. (2019) use a parameter study to investigate the performance effect of winglet length and cant angle for a small wind turbine and find that both power and thrust coefficients increase when a winglet is present. Using a low-speed wind tunnel

they fabricate and test six rotor configurations and are able to quantify the error ($\sim 5\%$) between their computational setup and the experimental data. Subsequently, an artificial neural network is used to predict the best winglet shape which has a 6% winglet length and a cant angle of $48°$ resulting in a 9% increase in both power and thrust coefficient.

     In summary, very few actual optimizations of wind turbine winglets have been found. While it in principle is fine to compare i) across model fidelity and ii) between gradient-based and gradient-free optimization procedures one can already identify one

major issue in doing so: The cited works rarely quantify how well the design optimization problem is solved and in not doing so one can basically not know if the problem is solved at all.

### 2.4   Overall trends in the covered literature

Tab. 1 should mainly be seen as an overview of numerical studies within wind energy of rotors fitted with winglets and correspondingly there are only a few studies (e.g., Tobin et al. (2015)) with experimental results whereas the remaining works

include numerical investigations. Turning to the nature of the cited studies one will find that 12 are shape analyses whereas only 6 include actual shape optimizations. It is fair to state that a chronological trend of transitioning from parameter studies to optimization studies can be observed since by far most optimization studies are very recent. For the parameter studies it is likely that better shapes could be found making it difficult to compare these with the optimization works. Still, it is possible to give several general statements which can be found below.



First of all there is a general consensus that winglets do indeed provide a promising means of increasing rotor performance. Some of the most studied effects are:

    - an increase in power production, (Lissaman and Gyatt, 1985; Imamura et al., 1998; Zahle et al., 2018; Hansen and Mühle, 2018; Sy et al., 2020),

    - noise reduction (Lissaman and Gyatt, 1985; Madsen and Fuglsang, 1997; Aravindkumar, 2014; Ebrahimi and Mar-
dani, 2018), and

    - accelerated wake recovery (Tobin et al., 2015; Kalken and Ceyhan, 2017; Aju et al., 2020; Mühle et al., 2020).

Starting with discussing the role of the winglet it is quite clear that there is a consensus in the literature. The underlying physical principle of a winglet is to mitigate the induced drag which is introduced on any loaded wing close to the tip where a spanwise velocity component flows from the pressure side to the suction side. This spanwise flow results in a tip vortex which
reduces the lift force. By tailoring the winglet one can manipulate the spanwise flow and change how and where the tip vortex occurs. The winglets role is therefore not necessarily in itself to increase power production locally but merely to transport the tip vortex further away to lower the induced drag which in turn raises the power production further inboard (Zahle et al., 2018; Hansen and Mühle, 2018; Sy et al., 2020; Mourad et al., 2020).

Also the direction of wind turbine winglets has been discussed in numerous studies. Several works (Zhu et al., 2017; Kha-
lafallah et al., 2019; Kulak et al., 2020) have found the best performance increase for upstream directed winglets. However, it has also been reported that downstream winglets are most efficient (Johansen and Sørensen, 2006; Bak et al., 2007; Gaunaa and Johansen, 2008; Ariffudin et al., 2016; Khalafallah et al., 2019) and that these should be relatively short (Bak et al., 2007). In the latter case it should be noted that even short winglets may raise concern on tower clearance (Johansen and Sørensen, 2006). Indeed, tower strike is an important aspect once transitioning from academic exercises to actual industrial applications.
The most straightforward conclusion to the above apparent contradiction on winglet direction is that it can be attributed to differences in parametrization and flow model fidelity. Furthermore, it is likely that the optimal downstream winglet and the optimal upstream winglet looks different and therefore should have different parametrizations.

Indeed, taking in the whole body of work mentioned above it can be seen that the parametrization of winglets and novel tip shapes vary greatly. Parametrizations of 2 to 12 design variables have been reported. In general, the most typical considered
design variables in Tab. 1 are winglet height, twist, sweep, cant angle, toe angle, extension and curvature. For the higher numbers reported, e.g. the 12 design variables used by Zahle et al. (2018) it is typical that the majority (9) of the design variables are used as planform-type design variables controlling e.g. the twist and chord distribution of the very tip of the blade. However, only three works (Hansen and Mühle, 2018; Zahle et al., 2018; Reddy et al., 2019) manage to take more than a few design variables into account simultaneously and it is fair to speculate whether a mere 2-4 design variables indeed are
sufficient to accurately model the full complexity of a winglet. That being said one can deduce several rules of thumbs for use of design variables: One should for example expect the optimizer to leverage flapwise displacement to decrease bending moment (Imamura et al., 1998) or manipulation of the thrust coefficient (Khalafallah et al., 2019). Flapwise displacement can also be found to be favoured slightly over sweep (Zahle et al., 2018, Fig. 4) but both are used heavily to displace the tip vortex in numerous works (Hansen and Mühle, 2018; Sy et al., 2020; Mourad et al., 2020). Few works also include twist and chord



design variables for the winglet planform where the expected trends of a reduced chord distribution towards the tip and an initial increase in twist distribution followed by a sharp decrease at the very end of the blade can be observed (Zahle et al., 2018). However, these trends for twist and chord probably are difficult to generalize and are likely to be specific to pressure side winglets.

Turning to the ability of the winglet to increase overall performance the results in Tab. 1 vary from an increase of 1.1 % to 20.1 % depending on objective function and parametrization. However, not all investigated shapes were reported to be load neutral compared to the baseline which could compromise already installed rotors. One should therefore only compare load neutral works (gray rows in Tab. 1) or clearly state by how much the initial load envelope is allowed to be exceeded in order to arrive at meaningful comparisons. Based on the load neutral works in Tab. 1 it is fair to state that only a few percent improvement can be expected depending on the choice of objective function.

Despite the above-mentioned salient features recent publications (Reddy et al., 2019; Papadopoulos et al., 2020) agree that winglet application still only has found a limited use in wind energy industry. Moving forward, at least two trends can be pointed out to help spread the role of the winglet and to counter remaining contradictions found in the literature:

- First action should be a general increased model fidelity when studying a feature such as the winglet. Numerous works (Matheswaran et al. (2019); Døssing (2007), etc.) point to the need of an increase in model fidelity when studying the tip area where conventional BEM models struggle. Using CFD and other high-fidelity reference also allows for improving said lower-fidelity models which has already been reported in several studies (e.g., Sørensen et al. (2011)). The increase in fidelity would also help better capture the complex flow phenomena and to better understand the role of the winglet.

- Secondly, one should aim for optimizations with unifying design optimization problems. By agreeing on unifying design optimization problems solved by the various frameworks one can eliminate some of the differences separating these works and more consensus should be possible.

## 3 Methodology

The overall aim of this study is to optimize the geometry of an academic wind turbine considering rotor aerodynamics only and all aeroelastic effects will be disregarded. Constraints on load and geometry are used as a surrogate to maintain structural feasibility. This section describes the overall design framework used in the study and remaining sections will describe the optimization problem in further detail as well as present the final results. Below, a general description of the framework is given (Sec. 3.1). Then, a description of the components; deformation library (Sec. 3.2), flow solver (Sec. 3.3), and optimizer (Sec. 3.4) is given.

### 3.1 FlowOpt: a high-fidelity shape optimization framework

The high-fidelity shape optimization framework called 'FlowOpt' at DTU Wind Energy is built around the in-house flow solver, EllipSys3D. The framework is focused on gradient-based shape optimization and in this study two different step-based approaches will be used to compute the gradient, namely the finite difference method and the Complex-Step method. The



Complex-Step method can provide machine accurate gradients and this method will therefore be used in a gradient verification of the finite difference method gradients. However, due to a lack of robustness in the current implementation of the Complex-Step method it is the finite difference method that will be used to compute the gradients during the actual optimizations.

All components in the FlowOpt framework are written in compiled low-level programming languages for maximum efficiency. However, these components also have user-friendly interfaces allowing for a lenient interaction using the interpreted high-level Python programming language. It is through these interfaces that the Python-based FlowOpt optimization framework is built. A visualization of the FlowOpt framework can be seen in Fig. 1.

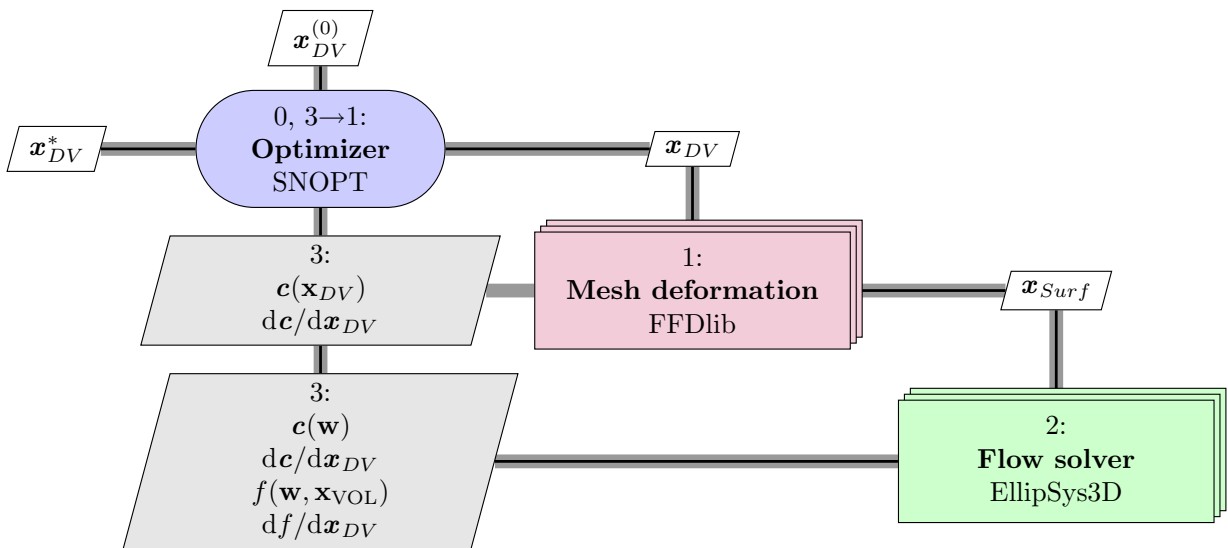

**Figure 1.** The FlowOpt framework when using a step-based approach for gradient computations.

The design framework visualization in Fig. 1 starts in the upper left corner where an optimizer (0, blue) sends a set of design
variables, $\mathbf{x}_{DV}$, to a mesh deformation library (1, red) which in turn sends a resulting deformed CFD mesh surface, $\mathbf{x}_{surf}$, to a flow solver (2, green) which finally can compute a flow field, $\mathbf{w}$, as well as functions of interest, $\mathbf{f}$, and constraints, $\mathbf{c}$. All functions of interest and constraints are then together with the related gradients returned to the optimizer (3, blue), and the overall cycle is repeated until a final design, $\mathbf{x}_{DV}^*$, is identified by the optimizer.

The choice of gradient computation method in the design framework has great impact on which optimization problems that
can be solved in a timely manner but the actual course an optimizer takes during an optimization should not change. Indeed, it has previously been shown (Madsen, 2020, Fig. 11.11) that the FlowOpt framework will carry out essentially identical optimizations when using either the Complex-Step method or the adjoint method as long as the flow field is well-converged to ensure that the gradients are machine accurate.

Returning to Fig. 1 it can be noted that there are several layers to the mesh deformation component (1, red) and flow solver
component (2, green). This signifies that the optimization may involve several simultaneous flow computations. As a result,



the design framework supports a nested parallelism (through OpenMDAO (Gray et al., 2019)) both with respect to running multipoint optimizations but also with respect to gradient evaluation. Thus, if sufficient computational resources are available one can dedicate a separate group of CPUs for each design variable and maintain a fixed gradient computation time. This allows for an execution time comparable to those seen for adjoint-based optimizations. However, unlike the adjoint method the cost in 490 computational resources will in this case increase linearly with the number of design variables being evaluated simultaneously. The presented study will therefore be rather costly in terms of CPU usage to ensure a state-of-the-art computation time.

### 3.2 FFDlib: a Free-form deformation library

The in-house Free-form deformation library, FFDlib (Madsen, 2020, Chapter 5), has been developed as an integral part of the FlowOpt design optimization framework and has previously been used in trailing edge flapping device studies (Horcas et al., 495 2018) as well as in high-fidelity shape optimization studies with various gradient computation techniques (adjoint, Complex-Step, etc.) as described elsewhere (Madsen, 2020). As the name suggests the parametrization library leverages a Free-form deformation (FFD) formulation to propagate changes from a few chosen design variables out to every single embedded mesh point in the computational mesh.

The FFD methodology has numerous salient features and was chosen particularly due to the following three aspects:

– exact numerical mesh representation (if the inverse search is converged to machine precision),

    – mesh topology agnostic, and

    – analytical gradients.

As explained in the original FFD paper (Sederberg and Parry, 1986) the basic principle in Free-form deformation is to embed an object in a rubber-like material meaning that for the present study the tip of the blade will be embedded in deformation 505 boxes. Mathematically speaking, an inverse search is used to map the discrete mesh points to the normalized parameter space spanned by the tuplet-coordinates, $(s,t,u)$, according to the following equation,

$$\mathbf{X}_{FFD}(s,t,u) = \sum_{i=0}^{l} \binom{l}{i}(1-s)^{l-i}s^i \left[ \sum_{j=0}^{m} \binom{m}{j}(1-t)^{m-j}t^j \left[ \sum_{k=0}^{n} \binom{n}{k}(1-u)^{n-k}u^k \cdot \mathbf{CP}_{ijk} \right] \right], \quad (1)$$

where $\mathbf{CP}_{ijk}$ are the $l \times m \times n$ control points of the FFD box and $\mathbf{X}_{FFD}(s,t,u)$ is the reconstructed 3-D point in the mesh computed from the normalized coordinates, $(s,t,u)$, and the control points, $\mathbf{CP}_{ijk}$.

The inverse search can be carried out by solving the equation (Casale and Stanton, 1985, eq. 7),

$$\begin{bmatrix} f_1 \\ f_2 \\ f_3 \end{bmatrix} = \mathbf{0} \Leftrightarrow \mathbf{X}_{FFD}(s,t,u) - \mathbf{P} = \mathbf{0}, \quad (2)$$



where $\mathbf{P}$ is the point which is to be embedded. This equation, is in FFDlib solved with the iterative Newton search:

$$
\begin{bmatrix} s \\ t \\ u \end{bmatrix}^{n+1} = \begin{bmatrix} s \\ t \\ u \end{bmatrix}^{n} - \begin{bmatrix} \frac{\partial f_1}{\partial s} & \frac{\partial f_1}{\partial t} & \frac{\partial f_1}{\partial u} \\ \frac{\partial f_2}{\partial s} & \frac{\partial f_2}{\partial t} & \frac{\partial f_2}{\partial u} \\ \frac{\partial f_3}{\partial s} & \frac{\partial f_3}{\partial t} & \frac{\partial f_3}{\partial u} \end{bmatrix}^{-1} \cdot \begin{bmatrix} f_1 \\ f_2 \\ f_3 \end{bmatrix}. \tag{3}
$$

The actual implementation of the above described FFD method is split in two parts to maximize efficiency: It consists of an
underlying code base written in Fortran containing the computationally heavy operations and a user-oriented interface written
in Python with which it is integrated in the FlowOpt shape optimization framework.

Furthermore, the basic FFD methodology can be extended in many ways such as ensuring $C^i$-continuity control between
deforming and non-deforming interfaces or volume preservation capabilities (see Hahmann et al. (2012)). While FFDlib has
indeed been extended with both mentioned features it is only the $C^i$-continuity control that will be used in the present study to
ensure a high mesh quality is maintained as described further in Sec. 5.1.

FFDlib will in the present study only be used to deform the CFD surface mesh. The deformed surface mesh will then be
propagated down to the flow solver which in turn updates the volume mesh.

### 3.3 EllipSys3D: a general purpose flow solver

For the present study the general purpose flow solver, EllipSys3D, based on the finite volume method is used to solve the
steady-state incompressible RANS equations as the underlying flow model along with the $\kappa - \omega$ shear-stress transport (SST)
turbulence model by Menter (1992). Furthermore, velocity and pressure variables are coupled using the semi-implicit method
for pressure-linked equations (SIMPLE) (Patankar, 1980) where the Rhie/Chow interpolation (Rhie and Chow, 1983) is used to
avoid checkerboard patterns. Flow solutions are obtained with a third-order accurate discretization scheme. Finally, the internal
volume mesh deformation routines will be used as explained further in Sec. 3.3.1.

At present, the EllipSys3D has been used extensively in numerous application areas ranging from blinded comparisons
(Simms et al., 2001), rotor analysis (Sørensen et al., 2002), DES simulations (Johansen et al., 2002), LES simulations (Berg
et al., 2018), and studies in vortex induced vibrations (Horcas et al., 2018) to name but a few. EllipSys3D was also recently
tested on the MareNostrum[4] supercomputer and achieved above a 50% scaling efficiency when using more than 16 thousand
CPUs. This aspect is particularly important for the present study where a high number of CPUs must be leveraged per rotor
computation in order to arrive at competitive computation timings for the entire optimization.

As seen, EllipSys3D has during the last three decades of development been thoroughly extended including an overset grid
method (Zahle, 2006), transition modelling (Sørensen, 2009), and an adjoint solver (Madsen, 2020). It is outside the scope of
the present work to account for all these applications. To be clear; the transition modelling was not used in the present study
and the flow was assumed to be fully turbulent. Alone within studies focused on tip shapes and winglets the flow solver has

---

[4]https://www.bsc.es/marenostrum/marenostrum, accessed Sep 30, 2021





been used in about a dozen works (Johansen and Sørensen, 2006, 2007; Johansen et al., 2008; Gaunaa and Johansen, 2007, 2008; Gaunaa et al., 2011; Sørensen et al., 2011; Kalken and Ceyhan, 2017; Zahle et al., 2018). Therefore, only a paragraph focusing on the shape optimization studies leveraging the EllipSys3D flow solver is given in the following.

The EllipSys flow solver has been used in slat design using an overset grid method (Gaunaa et al., 2013) on a 10 MW rotor configuration, where a series of 2D cross-sections were optimized with multi-element airfoils. EllipSys was also used to
design airfoils using a gradient-based setup with finite-differenced gradients, where the optimized airfoils were subsequently validated with wind-tunnel testing (Zahle et al., 2014). More recently, EllipSys3D figure in a surrogate-based design study (Zahle et al., 2018) where an optimization problem similar to the one seen in the present study was solved. They showed great promise in coupling the CFD solver to lower fidelity methods in the interest of saving computation time while maintaining the essential flow physics. Finally, the EllipSys3D flow solver have been integrated in the FlowOpt design optimization framework
and applied in high-fidelity shape optimization. The chosen design optimization problem (Madsen, 2020, Eq. 11.3) was a drag minimization of a 3-D wing subject to a lift constraint using five twist design variables. The problem was chosen to evaluate the newly developed framework on one of the few aerodynamic shape optimization problems that actually has an analytical optimal solution: an elliptic lift distribution. Mesh sizes up to $664 \cdot 10^3$ and $83 \cdot 10^3$ cells were used in the optimization using the Complex-step and adjoint method, respectively. It was found that the analytical elliptic lift distribution was well-approximated
(Madsen, 2020, Fig. 11.6 and 11.9) and most optimizations were tightly converged below a $10^{-4}$ threshold. Finally, it was demonstrated that essentially identical optimizations (Madsen, 2020, Fig. 11.11) occur when computing gradients either with the Complex-Step method or with the adjoint method for well-converged flows.

### 3.3.1 Mesh deformation

In this study the EllipSys3D flow solver will receive modified surface meshes as the optimization progresses. EllipSys3D will
then subsequently propagate the deformation change between the original surface mesh and the deformed surface mesh out through the volume mesh using internal mesh deformation propagation routines. This deformation method in EllipSys is based on an analytical approach in which both the translatoric and overall re-orientation (i.e., rotation) of the surface are propagated and attenuated in the volume mesh using a hyperbolic tangent function, blended into the original volume mesh based on the distance to the blade surface along the given grid line. The consideration of the rotation of the surface greatly improves the
quality of the deformed meshes for certain configurations compared to attenuating only the displacement resulting from the deformation. For instance this update has been instrumental for the present work, as considerable local rotations were identified for some of the explored tip shapes. Without the consideration of rotations in the mesh deformation routines, this led to mesh folding in the boundary layer region.





### 3.4 Optimizer

In this work the Sparse Nonlinear OPTimizer (SNOPT) version 7.2-10 is used (Gill et al., 2018, 2005)[5]. SNOPT is based on a sequential quadratic programming (SQP) algorithm which allows for infeasible steps to be taken during the optimization. At convergence all optimizations were feasible within the requested tolerance.

The SNOPT optimizer is accessed in the FlowOpt framework through the open-source Python wrapper called pyOptSparse (Wu et al., 2020) by choosing the `pyOptSparseDriver` in OpenMDAO. pyOptSparse is leveraged extensively[6] throughout
the broader numerical optimization community and is as the name suggests dedicated to constrained nonlinear optimization of large sparse problems.

### 4 Baseline analysis

The present section contains a description of baseline planform, surface mesh and volume mesh. Mesh modifications were made to the original geometry to enable a deep convergence. These changes will also be presented and the effects of the
geometry changes will be assessed. Finally, a scaling study is also included to discuss the necessary computational resources needed to carry out direct CFD-based optimizations.

### 4.1 Computational mesh

The baseline geometry is the IEA 10 MW reference wind turbine[7][8] (Bortolotti et al., 2019, Appendix B). The chord and twist planform distributions can be inspected in Fig. 2.

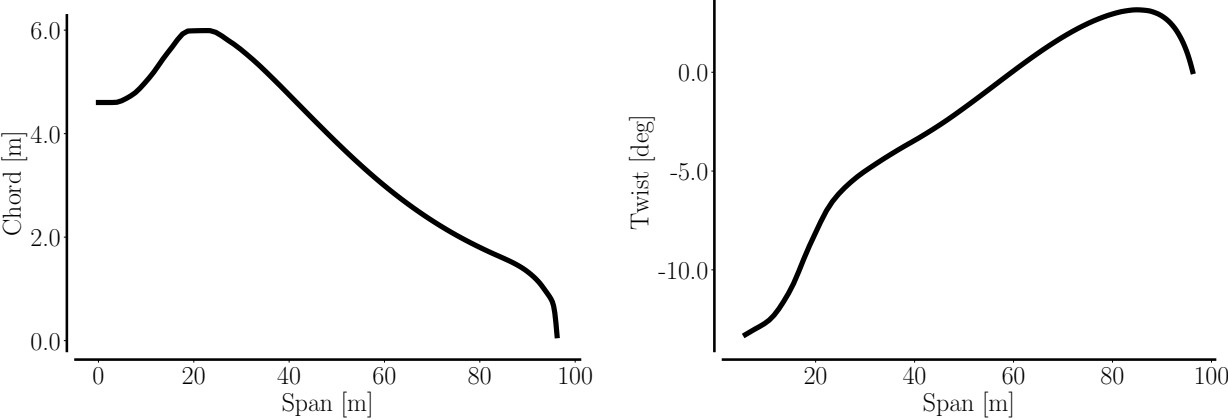

**Figure 2.** The IEA 10 MW wind turbine (baseline) planform.

---

[5]https://web.stanford.edu/group/SOL/guides/sndoc7.pdf, accessed Sep 30, 2021
[6]https://mdolab-pyoptsparse.readthedocs-hosted.com/en/latest/publishedWorks.html, accessed Sep 30, 2021
[7]https://www.nrel.gov/docs/fy19osti/73492.pdf, accessed Sep 30, 2021
[8]https://github.com/ieawindtask37/iea-10.0-198-rwt, accessed Sep 30, 2021





Now follows a description of how the structured surface and volume meshes are generated. Both surface mesh and volume mesh can be inspected in Fig. 3.

**Figure 3.** Surface (upper, left) and volume (upper, right) where a close-up one of the blade meshes is given below. Visualization of tip cap mesh (lower, left) and a view of the airfoil discretization is also offered (lower, right). Notice, that for clarity only 1 out of 4 mesh lines is shown. Inflow (transparant gray) and outflow (gray with black mesh lines) zones are visualized in the upper right plot.

The rotor surface mesh has been generated from the planform data and the FFA-W3 airfoil family used on the IEA 10 MW with the in-house Parametric Geometry Library (PGL) tool. The surface mesh on each blade has 256 cells in the chordwise direction and 128 cells in the spanwise direction partitoned into blocks of $32 \times 32$ cells. Four blocks of $32 \times 32$ cells form the

tip cap resulting in a total of $3 \cdot 36 \cdot 32 \cdot 32 = 110592$ mesh cells for the surface mesh.

The baseline volume mesh is prepared with the in-house hyperbolic mesh generator, HypGrid3D (Sørensen, 1998). The volume mesh has an O-O topology where 128 layers are grown from the surface mesh resulting in $14.16$ million cells. By setting the first boundary layer cell below $10^{-6}$m a $y^+$ below 1.0 is ensured given the operational conditions seen in Tab. 2.





The above description is for the baseline mesh at the very start of the optimization. All subsequent volume meshes throughout
the optimization are computed using the internal mesh deformation routines in EllipSys3D. As a result, they are all likely to
exhibit a slight reduction in mesh quality due to impaired orthogonality of the mesh as the tip is created from a straight blade
planform.

Finally, with respect to boundary conditions the rotor surface is a no-slip boundary whereas the farfield zone is split into two
sections: An approximately circular area behind the rotor is an outflow-scaling zone whereas the rest of the farfield region is a
(uniform) inflow zone.

## 4.2   Geometrical modifications

Initially, the described surface mesh of IEA 10 MW reference wind turbine was used in the optimizations without any additional
modifications. However, as finer grid levels were taken into use a seriously impaired gradient quality was observed. The
explanation proved to be a lack of flow convergence on the finer grid levels due to complex swirling 3-D flow phenomena
occurring in the root area of the rotor. Such a massive blunt object will inherently cause complex flow phenomena resulting in
lack of convergence for steady-state incompressible CFD solvers based on the SIMPLE algorithm.

There are several remedies suggested in wind energy research on shape optimization to the described convergence issues.
Nielsen and Diskin (2012) were able to carry out shape optimization not just of the rotor itself[9] but also including nacelle
and tower (Nielsen and Diskin, 2012, Fig. 4) by using an unsteady RANS formulation. It is very likely that the unsteady
RANS formulation in EllipSys3D would somewhat mitigate the observed impaired convergence. However, the computation
time would drastically increase which is why this option was ruled out.

Yet another option found in the literature is to exclude some mesh regions: Dhert et al. (2017) cut out the root of the rotor
configuration to improve convergence whereas Vorspel et al. (2018) exclude both root and tip regions from being deformed
due to a local inferior gradient quality.

Of the above mentioned alternatives the mesh (root) modification option is favoured since the present shape optimization
study is focused on tip shapes. However, instead of removing the root section altogether it was decided to simply re-shape it
aerodynamically to reduce separation and in turn improve convergence. The modified baseline mesh can be inspected in Fig. 4.

To assess the quality of the modified mesh a full grid sequence is run on both the original baseline mesh and the modified
baseline mesh using the operational conditions seen in Tab. 2.


The resulting convergence behaviour as well as spanwise forces can be inspected in Fig. 5. As is evident from the figure it
is only the innermost part of the spanwise forces where one can discern a minor change. The design optimization problem of
optimizing the tip is in other words practically unaltered and the convergence issues have been addressed without changing the
task at hand.

---

[9]Interested readers will notice that the actual root of the NREL VI rotor was excluded in this study. Still, the inclusion of nacelle and tower introduce a
massive blunt object in the flow



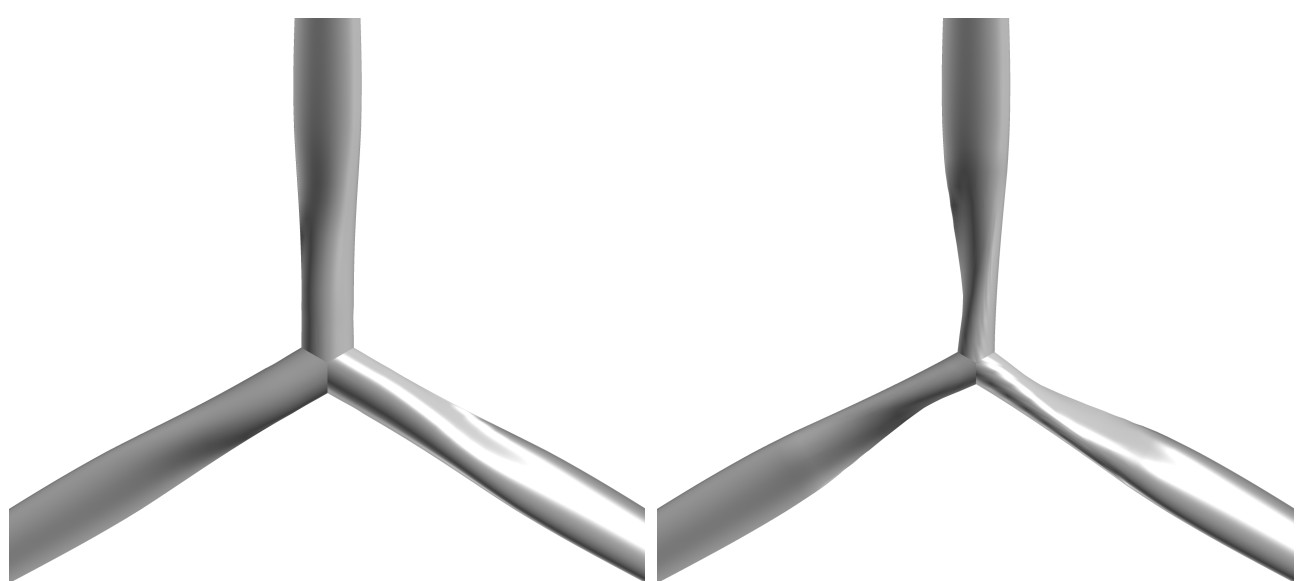

**Figure 4.** The IEA 10 MW reference wind turbine baseline geometry (left) and needed mesh modifications (right) resulting in an improved flow solver convergence.

**Table 2.** Operational conditions for the simulations used in both analysis and optimization. Density is set to the density of air at sea level and 15 °C, $\rho = 1.225 \ [kg \ m^{-3}]$ and dynamic viscosity is set to $\mu = 1.784 \cdot 10^{-5} \ [kg \ m^{-1} \ s^{-1}]$.

| Run | Wind speed | RPM | Rotation rate, $\omega$ | Pitch |
|-----|-----------|-----|------------------------|-------|
| | $[m/s]$ | $[-]$ | $[rad/s]$ | $[deg]$ |
| wsp08 | 8.0 | 8.164590 | 0.8681 | 0.0 |

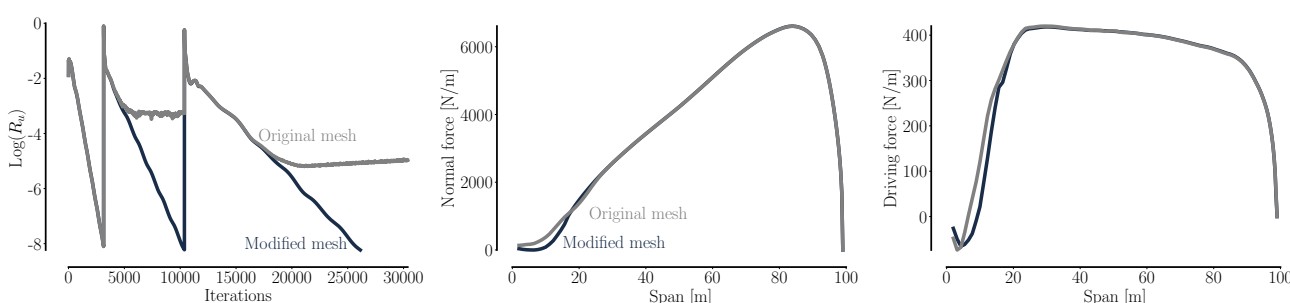

**Figure 5.** A comparison of the baseline mesh before (gray) and after (black) root modifications reveal the modified baseline mesh ensures a deep convergence (left) on all the grid levels; L3 (it=0-4000), L2 (it=4000-12000) and L1 (it=12000-30000). Mesh levels are listed in Tab. 3. Also the normal force (middle) and driving force (right) is visualized. As seen, the spanwise forces only differ noticeably within the 30 m farthest inboard.



## 4.3 Mesh convergence study

Based on the mesh of the modified rotor geometry a mesh convergence study is carried out to investigate the grid dependence of the flow solution. To generate the coarse mesh levels to be used in the mesh convergence study a grid coarsening sequence inside the flow solver is used: The above described computational mesh is the finest mesh called L1. The next mesh level, L2, is generated by removing every second mesh point in all directions. Similarly, the L3 mesh level, which is the coarses mesh level used in the present study is generated by removing every second grid mesh point from L2. All three mesh levels, L1, L2, and L3, are listed in Tab. 3.

It has previously been shown (Madsen et al., 2019, Tab. 4) that a flow solver with a noticeable grid dependency may even suggest wrong design trends (Madsen et al., 2019, Sec. 6.2) on the coarser grid levels. Therefore, one should ensure that all mesh levels exhibit low error percentages compared to the Richardson extrapolation. As listed in Tab. 3 the error percentage on both mesh level L1 and L2 are well below $10\%$ and it should be reasonable to expect relevant design optimization results at least for these two mesh levels.

| Mesh | Cells | QUICK: Third order stencil | | | |
| | | Torque | Error | Thrust | Error |
| | [million] | $\cdot 10^6$[Nm] | [%] | $\cdot 10^6$[N] | [%] |
|---|---|---|---|---|---|
| L3 | 0.221 | 5.676 | 10.5 | 1.169 | 6.2 |
| L2 | 1.769 | 5.394 | 5.0 | 1.124 | 2.1 |
| L1 | 14.155 | 5.200 | 1.3 | 1.106 | 0.5 |
| Extrapolation | $\infty$ | 5.136 | 0.00 | 1.101 | 0.0 |

**Table 3.** Mesh convergence study with error percentages obtained from Richardson extrapolations. The quadratic upstream interpolation for convection kinematics (QUICK) discretization scheme is used for all cases.

## 4.4 Computational resources

As a final section in the baseline analysis the computational resources needed to carry out shape optimizations on all three grid levels are assessed. A visualization of flow solver scaling on the HPC cluster[10] at DTU Wind Energy can be inspected in Fig. 6.

The most important aspect to consider when inspecting scaling results as shown in Fig. 6 is the finest mesh level, L1, since it by far is the most time consuming and since it will determine the final results. As seen, the scaling is indeed close to ideal on L1 for EllipSys3D and if computational resources are available one can advantageously use as many CPUs as possible on the given rotor mesh. In the present case that number is 432 - one CPU for each block in the mesh.

---

[10]The Sophia HPC cluster at DTU Wind Energy comprises 516 computational nodes where each of these is a x86-64 computer with 32 cores. More information is available at https://windenergy.dtu.dk/nyheder/2019/12/ny-computer-cluster-paa-risoe-campus?id=a495d2e5-a7f1-4133-9fb2-488d150f7c01, accessed Sep 30, 2021



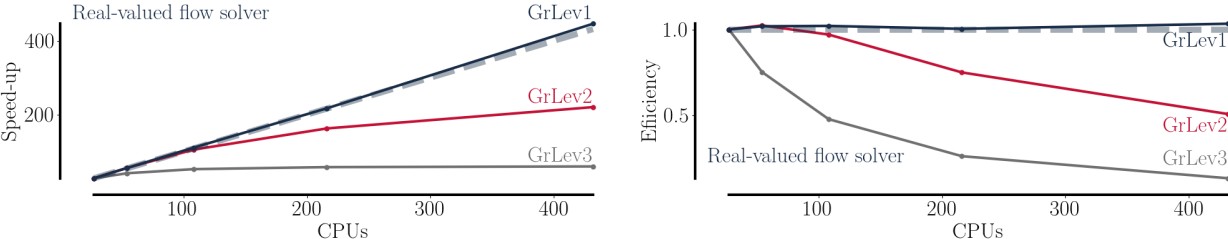

**Figure 6.** Measured scaling on the Sophia cluster for the EllipSys3D flow solver on the modified baseline mesh using 27, 54, 108, 216, and 432 CPUs, respectively. For the L2 mesh it seems up to 108 CPUs will result in very efficient CPU usage. For L1 which is by far the most interesting and time consuming mesh level the scaling is ideal and up to 432 CPUs can advantageously be used. On L3 the mesh is so coarse that the computational task is modest making scaling on this mesh level less important.

## 5 Design optimization problem

The singlepoint design optimization problem used in the present study is to optimize the power production using 12 design variables while satisfying constraints on smoothness of the geometry and on the flapwise bending moment computed at 90 % span, which from now on will be written as 0.9 r/R for brevity. Mathematically, the design optimization problem can be formulated as:

$$
\begin{aligned}
\text{minimize:} \quad & -\frac{P(\mathbf{x})}{P(\mathbf{0})} \\
\text{with respect to:} \quad & \text{twist: } \theta_1, \theta_2, \theta_3, \theta_4 \\
& \text{chord: } c_1, c_2, c_3, c_4 \\
& \text{tip shape: } c_{wl}, h_{wl}, s_{wl}, R_{ext} \\
\text{subject to:} \quad & \frac{M_{\text{bending}}\big|_{0.9_{r/R}}}{M_{\text{bending}}\big|_{0.9_{r/R}\ \text{initial}}} \leq 1., \\
& \frac{\mathrm{d}chord}{\mathrm{d}S} \leq 0.
\end{aligned}
\tag{4}
$$

Above, the objective function is mechanical power, $P = \omega \cdot Q$, found from rotational rate, $\omega$, and torque, $Q$, where the operational conditions used in this study can be found in Tab. 2.

As seen, the constraint on bending moment ensures that the bending moment at 0.9 r/R span does not increase compared to the baseline value. The geometric chord design variable constraint ensures that the optimization does not increase the chord towards the very tip of the blade. There are no constraints on the twist design variables. In the following section (Sec. 5.1) the

660 12 design variables are further described.



## 5.1 Parameterization

In the design optimization problem there are 12 design variables: 4 twist variables, 4 chord variables, 1 extension variable ($R_{ext}$), 1 flapwise tip displacement variable ($h_{wl}$), 1 edgewise tip displacement variable ($s_{wl}$), and 1 tip curvature variable ($c_{wl}$). We have for the present study adopted all design variables nomenclature from a related surrogate-based optimization study (Zahle et al., 2018) to allow for an easy comparison. The twist, chord, and extension design variables change the planform at the blade tip region whereas $c_{wl}$, $h_{wl}$, and $s_{wl}$ are used to shape the tip towards winglet-like shapes.

All 12 design variables are imposed on the CFD mesh by the optimizer by manipulating the three FFDlib tip boxes (blue) seen in Fig. 7. As seen, the deforming tip region (red mesh lines) embedded in the FFD boxes only make up 10% of the total blade length of the baseline rotor (gray). Also seen are two darker blue areas on the FFD boxes signifying that the related FFD spanwise sections have a particular functionality: The darker blue part further inboard visualize the two FFD box sections which are locked to ensure a $C^1$-continuity with the remaining surface mesh. The darker blue part towards the very tip of the blade visualize three FFD sections that have been inserted as a tip-cap protection since it is crucial that this part of the mesh retains a high mesh quality. Notice, that the tip can indeed still be deformed by moving the outermost FFD box section in which case the three tip-cap protection sections are interpolated to their correct position.

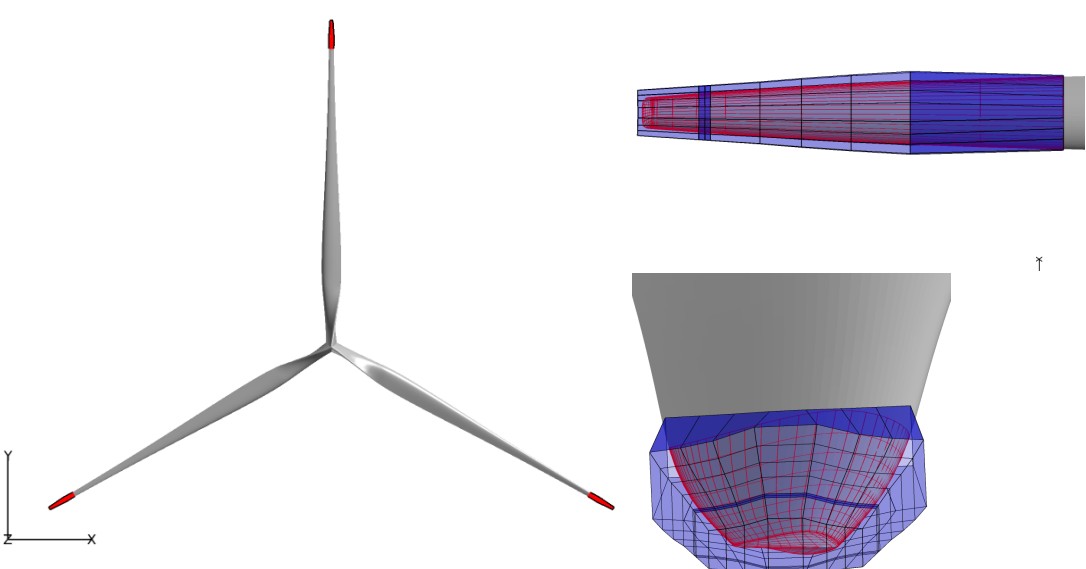

**Figure 7.** Overview of baseline rotor (gray), deformable tip region (red), and FFD boxes (blue). Grid level 3 is used to visualize mesh lines.





### 5.1.1 Tip design variables

The three tip design variables: flapwise tip displacement ($h_{wl}$), edgewise tip displacement ($s_{wl}$), and curvature of the tip ($c_{wl}$) can be inspected in Fig. 8 where the FFD box visualization has been omitted in order to more easily inspect the deformed geometry. Notice, that the curvature is an interpolation variable from 0.0 (max. curv.) to 1.0 (straight tip). It is also relevant to point out, that the role of the three locked FFD sections (dark blue in Fig. 7) next to the outermost tip section is to protect the volume cells at the tip which at the same time results in a reduced maximal curvature for the parametrization.

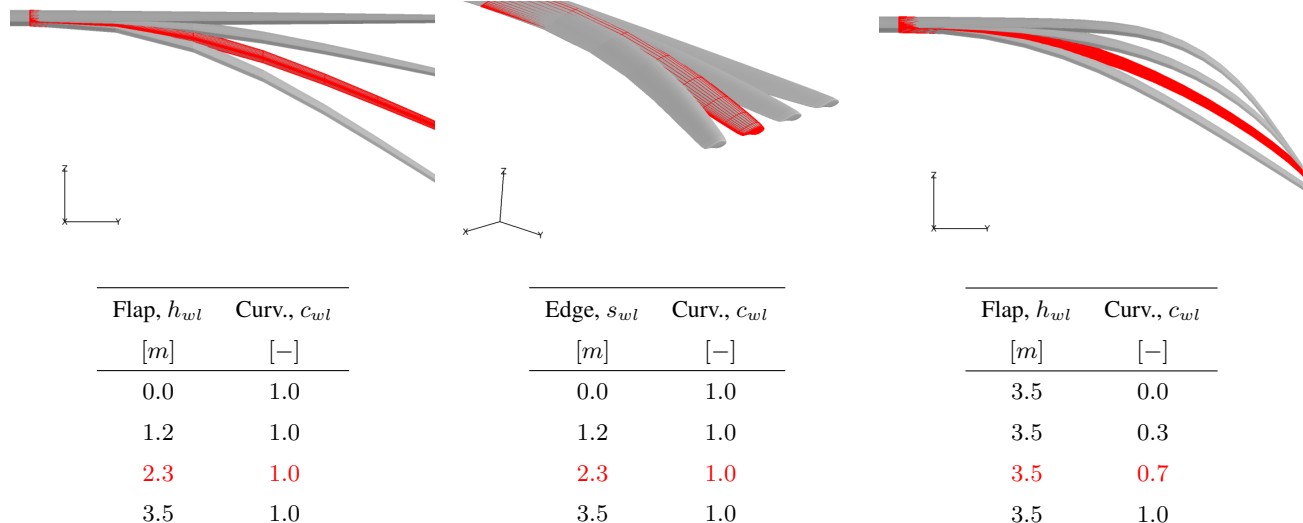

| Flap, $h_{wl}$ | Curv., $c_{wl}$ | | Edge, $s_{wl}$ | Curv., $c_{wl}$ | | Flap, $h_{wl}$ | Curv., $c_{wl}$ |
| --- | --- | --- | --- | --- | --- | --- | --- |
| [m] | [−] | | [m] | [−] | | [m] | [−] |
| 0.0 | 1.0 | | 0.0 | 1.0 | | 3.5 | 0.0 |
| 1.2 | 1.0 | | 1.2 | 1.0 | | 3.5 | 0.3 |
| 2.3 | 1.0 | | 2.3 | 1.0 | | 3.5 | 0.7 |
| 3.5 | 1.0 | | 3.5 | 1.0 | | 3.5 | 1.0 |

**Figure 8.** Tip design variables: flapwise tip displacement (left), edgewise tip displacement (middle), and curvature of the tip (right). The used design variables are listed below each visualization. The row colored in red show the design variable settings used to generate the red meshes. The flapwise and edgewise displacement visualizations are made with no curvature forcing the tip to be as straight as possible. The curvature visualization is made with a maximally flapped tip. Grid level 3 is used to visualize mesh lines.

## 5.2 Gradient verification

Given that the present study uses the finite difference method an initial step size study is carried out to identify a suitable step size. Tab. 4 shows how the finite difference gradient accuracy correlates to the chosen finite difference step size. The machine accurate reference gradient is computed using the Complex-Step method. Gradient accuracy for all constraints have also been verified (not shown) and exhibit similar accuracy.

By inspecting Tab. 4 one learns that for most design variables (e.g., twist, edge, extension) the best finite difference step size is $10^{-4}$. However, for chord design variables a step size of $h = 10^{-5}$ seems better suited. Up to 6 significant digits can be seen which is more than plenty to carry out gradient-based design optimization. However, it is also evident that the step size does not have to be much off before the gradient precision is worsened. This may factor in for longer optimizations where many





**Table 4.** Table showing how the finite difference gradient accuracy depends on the chosen step size. The machine accurate reference gradient is computed with the Complex-Step method.

| Step size, $h$ | Significant digits on L3: | | | |
|---|---|---|---|---|
| | Twist, $\theta_1$ ($\cdot 10^{-4}$) | Twist, $\theta_2$ ($\cdot 10^{-4}$) | Twist, $\theta_3$ ($\cdot 10^{-4}$) | Twist, $\theta_4$ ($\cdot 10^{-4}$) |
| $10^{-1}$ | $-5.85049867$ | $-4.43009974$ | $-3.34042748$ | $-6.96420069$ |
| $10^{-2}$ | $-5.86822272$ | $-4.44143106$ | $-3.34801787$ | $-7.04257508$ |
| $10^{-3}$ | $-5.87003860$ | $-4.44256502$ | $-3.34877921$ | $-7.05052147$ |
| $10^{-4}$ | $-5.87025903$ | $-4.44271548$ | $-3.34889474$ | $-7.05135166$ |
| $10^{-5}$ | $-5.87069215$ | $-4.44313697$ | $-3.34930883$ | $-7.05186998$ |
| $10^{-6}$ | $-5.87511817$ | $-4.44770443$ | $-3.35383055$ | $-7.05632885$ |
| (reference) | $-5.87055619$ | $-4.44270932$ | $-3.34887356$ | $-7.05133069$ |
| | Chord, $c_1$ ($\cdot 10^{-3}$) | Chord, $c_2$ ($\cdot 10^{-3}$) | Chord, $c_3$ ($\cdot 10^{-3}$) | Chord, $c_4$ ($\cdot 10^{-2}$) |
| $10^{-1}$ | $-8.79500858$ | $-6.08880387$ | $-4.32113105$ | $-1.03168014$ |
| $10^{-2}$ | $-8.12767663$ | $-5.70326167$ | $-4.06860284$ | $-0.80373667$ |
| $10^{-3}$ | $-8.06665832$ | $-5.66956278$ | $-4.04640816$ | $-0.78776642$ |
| $10^{-4}$ | $-8.06085604$ | $-5.66622427$ | $-4.04407679$ | $-0.78611997$ |
| $10^{-5}$ | $-8.06021495$ | $-5.66583677$ | $-4.04381366$ | $-0.78595010$ |
| $10^{-6}$ | $-8.05994610$ | $-5.66558572$ | $-4.04358425$ | $-0.78591230$ |
| (reference) | $-8.06008832$ | $-5.66584359$ | $-4.04382519$ | $-0.78592965$ |
| | Curv., $c_{wl}$ ($\cdot 10^{-3}$) | Flap, $h_{wl}$ ($\cdot 10^{-4}$) | Edge, $s_{wl}$ ($\cdot 10^{-4}$) | Ext., $R_{ext}$ ($\cdot 10^{-1}$) |
| $10^{-1}$ | $-3.74767833$ | $+3.28947962$ | $-3.19416366$ | $-1.41397308$ |
| $10^{-2}$ | $-3.86239738$ | $+3.64583550$ | $-3.71786342$ | $-1.41788435$ |
| $10^{-3}$ | $-3.87348172$ | $+3.67988163$ | $-3.77135090$ | $-1.41846530$ |
| $10^{-4}$ | $-3.87460335$ | $+3.68325048$ | $-3.77628964$ | $-1.41852803$ |
| $10^{-5}$ | $-3.87475825$ | $+3.68314301$ | $-3.77729514$ | $-1.41853454$ |
| $10^{-6}$ | $-3.87520704$ | $+3.67891606$ | $-3.78163278$ | $-1.41853715$ |
| (reference) | $-3.87632872$ | $+3.68048836$ | $-3.77670622$ | $-1.41852678$ |

Red digits (i.e., $\theta_4 [h = 10^{-1}]$, $c_1 [h = 10^{-6}]$, and $h_{wl} [h = 10^{-3}]$) are used to indicate that the error is not on the indicated digit, but one order of magnitude below.





new rotor shapes are introduced meaning that also the optimal step size might change slightly throughout the course of the optimization.

## 6 Results

The main results of this study fall in two parts. First part (Sec. 6.1) is carried out solely on mesh level L3 and is a study in finding the best settings that balance most accurate gradient computation on one hand with more robust settings on the other
hand that result in properly converged optimization problems. The second part (Sec. 6.2) is a shape optimization study using three grid levels where focus is laid on analyzing the shape of the resulting blade tip.

### 6.1 Step size study for shape optimizations based on finite difference

To identify the best functioning step size a series of six shape optimizations were run using step sizes $h = 10^{-1}$, $h = 10^{-2}$, $h = 10^{-3}$, $h = 10^{-4}$, $h = 10^{-5}$, and $h = 10^{-6}$. All optimizations were allowed to run until either the upper limit of 100 major
iterations (i.e., design steps) were reached or the optimizer exited since it could not converge the design problem further[11]. The lower and upper design variable bounds for this step size study are:

$$\begin{bmatrix} -10.0, -10.0, -10.0, -10.0 \\ \cdot 0.5, \ \cdot 0.5, \ \cdot 0.5, \ \cdot 0.5 \\ \cdot 0.0, \ + 0.0, +0.0, \ \cdot 1.0 \end{bmatrix} \leq \begin{bmatrix} \theta_1, \ \theta_2, \ \theta_3, \ \theta_4 \\ c_1, \ c_2, \ c_3, \ c_4 \\ c_{wl}, h_{wl}, s_{wl}, R_{ext} \end{bmatrix} \leq \begin{bmatrix} +5.0, \ +5.0, \ +5.0, \ +5.0 \\ \cdot 1.0, \ \cdot 1.0, \ \cdot 1.0, \ \cdot 1.0 \\ \cdot 1.0, \ +3.5, \ +3.5, \ \cdot 2.0 \end{bmatrix} \qquad (5)$$

Above, the units for the design variables are the following: Twist variables, $\theta$, are in degrees. Chord variables, $c$, use a unit less scaling factor. Curvature is likewise a unit less interpolation factor from 0.0 to 1.0 where 0.0 represents the maximally
allowed curvature for the parametrization and 1.0 represents a straight tip (see Fig. 8, right). Flapwise displacement, $h_{wl}$, and edgewise displacement, $s_{wl}$, are in meters and the extension scaling variable, $R_{ext}$, is a unit less scaling variable which stretches the entire FFD box.

The shape optimization results for the six different step sizes are listed in Tab. 5 and their optimization histories are visualized in Fig. 9. Before discussing the results in Tab. 5 it should be noted that the step size is but one of several important settings that
one must fine tune to arrive at a properly functioning optimization framework. Another important consideration to mention is that these step size studies ideally should be done on each grid level. However, that is extremely expensive on the finest grid level for 12 design variables. Based on the present study it is the authors' experience that for flow solvers as consistent across grid levels as seen in Tab. 3 it will suffice to carry out the step size study on mesh level L3. For flow solvers less consistent across grid levels one might have to redo the step size study on each grid level.

The first thing to note when inspecting Tab. 5 is that one should take care not to use too small finite difference step sizes. Indeed, the most accurate gradient step sizes identified in Sec. 5.2 (i.e., $h = 10^{-4}$ and $h = 10^{-5}$) do not result in very successful optimizations. Although $h = 10^{-4}$ and $h = 10^{-5}$ indeed where the most accurate step sizes on the baseline mesh they do not

---

[11]The related SNOPT message is: SNOPTC EXIT 40 – terminated after numerical difficulties. SNOPTC INFO 41 – current point cannot be improved



**Table 5.** Overview of six CFD-based shape optimizations of wind turbine blade tips using different finite difference step sizes. Operational conditions are found in Tab. 2. Naming convention explanation using **SPL3e2c** as an example:

(**SP**:) a singlepoint optimization (**L3**:) on mesh level L3 (**e2**:) using a finite difference step size of $10^{-2}$ (**c**:) (cold-)started from a straight baseline configuration.

All optimizations were carried out using 648 CPUs split into 12 groups - one group of 54 CPUs for each design variable.

| ID | Mesh level | Wall clock⋆ | Maj. iter. | Step size | Convergence | Mech.Power |
|---|---|---|---|---|---|---|
| | (See Tab. 3) | $10^{-7}$ threshold / full [h] | | (h=) | (Orders of Magn.) | (improvement) |
| SPL3e6c | L3 | - / 28.2 | 88 | $10^{-6}$ | 1 | 0.37 % |
| SPL3e5c | L3 | - / 14.1 | 12 | $10^{-5}$ | 1 | 0.37 % |
| SPL3e4c | L3 | - / 11.6 | 40 | $10^{-4}$ | 2 | 0.39 % |
| SPL3e3c | L3 | 6.0 / 11.6 | 54 | $10^{-3}$ | 8 | 0.39 % |
| SPL3e2c | L3 | 5.3 / 10.5 | 46 | $10^{-2}$ | 8 | 0.39 % |
| SPL3e1c | L3 | 5.7 / 9.2 | 50 | $10^{-1}$ | 9 | 0.39 % |

⋆ Given that some groups of CPUs on the HPC cluster will be faster than other CPU groups a representative computation speed ([it]/[sec]) for each grid level has been computed as an average over an entire optimization on a given grid level. This allows for a fair comparison between optimizations carried out on the same grid level although they have not been computed using the exact same CPUs.

(in the present study) seem to be a robust choice over the course of an entire optimization. One potential explanation could be that cancellation errors due to too small step sizes are likely to occur for at least a few of the several hundred rotor shapes generated by the optimizer during an optimization. Even if this happens for just a few shapes it might result in the optimizer having to reset the Hessian which could impair the optimization problem convergence. One could implement individual step sizes for each design variable in the attempt to gain better gradient accuracy and robustness but in general it seems advisable to use a step size 1-2 orders of magnitude larger than $h = 10^{-4}$ to deeply converge optimization problems.

Judging from Tab. 5 the main benefit from finding a good compromise between gradient accuracy and overall robustness is that a sound procedure for ending the optimization is available: Optimizations with a functioning step size (SPL3e1c, SPL3e2c, and SPL3e3c) all have around 50 major iterations when terminating and result in approximately the same improvement whereas the other optimizations (SPL3e4c, SPL3e5c, SPL3e6c) differ both in number of major iterations as well as on the final improvement. Noticeably, these optimizations (e.g., SPL3e4c) may still result in approximately as much improvement as the better performing optimizations but the optimization problem is only converged 1-2 orders of magnitude and one cannot be sure whether the optimization problem is actually solved.

Optimizations SPL3e5c and SPL3e6 in Tab. 5 resemble two typical end case scenarios for optimizations with inefficient settings. Either the optimization finishes prematurely (e.g., SPL3e5c) or the optimizer keeps trying to converge the optimization problem using excessive iterations (e.g., SPL3e6c). In both cases, the optimizer exits due to numerical difficulties resulting from the inaccurate gradient. While the SPL3e5c optimization finishes much sooner than the SPL3e6c optimization thus saving





considerable computation time it actually results in a shape which performs slightly worse (not visible from Tab. 5). Thus, SPL3e5c represents the least successful optimization in Tab. 5.

Turning to Fig. 9 it is easy to see how the optimizations progress. All optimization problems start with an optimality around $10^{-3}$ (see right y-axis in Fig. 9 shown in red). The $10^{-7}$ threshold has been chosen to ensure that all design optimization problems are converged about four orders of magnitude. To relate this to relevant literature it is noted that a wind turbine

design problem with a single design variable (pitch) has been well-converged with an optimality reduction of only three orders of magnitude on similar mesh sizes ($325 \cdot 10^3$ cells in (Dhert et al., 2017, Fig. 5) and $221 \cdot 10^3$ cells in (Madsen et al., 2019, Fig. 8)). Indeed, Madsen et al. showed that wind energy design optimization problems with 1, 14, and 154 design variables were all well-converged with an optimality reduction of about 2-4 orders of magnitude (Madsen et al., 2019, Fig. 8, 9, 11, 12, 14, and 17). Therefore, a chosen threshold of about four orders of magnitude reduction in optimality for the present study

seems reasonable. As seen in Fig. 9 the chosen threshold is only reached by the optimizations which converge properly and the threshold could even have been 3-4 orders of magnitude stricter at the limited expense of about 10 major iterations. However, it is clear from the merit function shown in black that it would lead to no added value since the objective function for all practical purposes is fully converged. The optimization history for SPL3e6c clearly visualizes what may happen if said threshold is not met: The optimizer uses a large amount of excessive iterations in the attempt to further converge the optimization problem.

Evidently, it is crucial to have a well-defined way to end optimizations.

An important point when discussing the cost of optimizations is to discern an optimizer's major iterations from wall clock computation time: Although SPL3e5c is the optimization with by far fewest major iterations it is the second most wall clock time consuming optimization as also indicated in Tab. 5. The optimizer simply takes very few actual steps in this optimization due to impaired gradient precision.

In summary, the best functioning step sizes seem to be $h = 10^{-1}$, $h = 10^{-2}$, and $h = 10^{-3}$. Given that $h = 10^{-3}$ of the three resulted in the most accurate gradient computation when comparing to machine accurate reference gradients (Tab. 4) it is the $h = 10^{-3}$ step size that will be used in the ensuing section.

### 6.2 Aerodynamic shape optimization of wind turbine blade tips

With $h = 10^{-3}$ identified as a promising finite difference step size in Sec. 6.1 a shape optimization study across three grid

levels has been carried out. All optimizations were again allowed to run until either the upper limit of 100 major iterations (i.e., design steps) were reached or the optimizer exited since it could not converge the design problem further[12]. For large flapwise and edgewise displacements negative cell volumes were encountered on the finest mesh level as a result of the deformation of the mesh during the optimization. In order to make sure that the exact same optimization could be carried out on the various grid levels it was therefore necessary to limit the upper design variable bounds for $h_{wl}$ and $s_{wl}$ to 2.0[m]. All other settings

from Sec. 6.1 were kept the same. The results from the final shape optimization study are listed in Tab. 6 and their optimization histories are visualized in Fig. 10.

---

[12]The related SNOPT message is: `SNOPTC EXIT 40 - terminated after numerical difficulties. SNOPTC INFO 41 - current point cannot be improved`

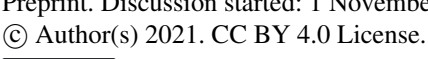



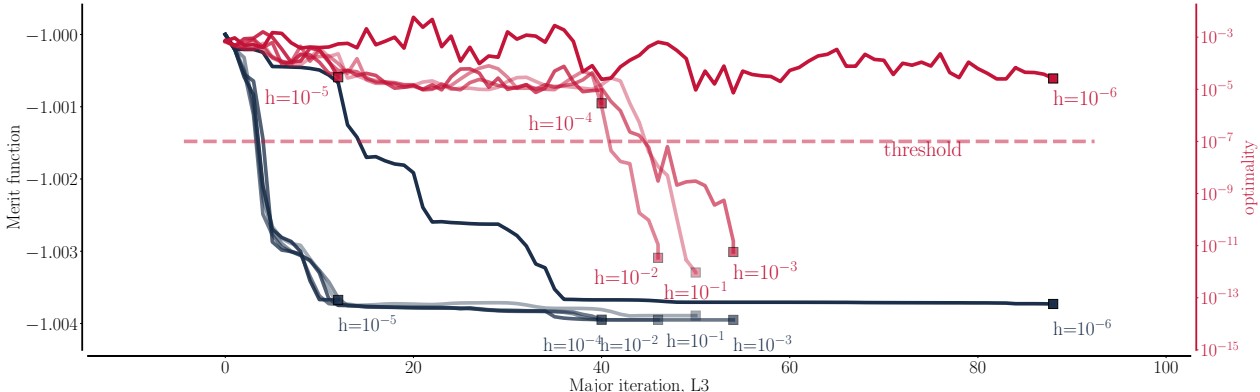

**Figure 9.** Merit function (black, left y-axis) and Optimality (red, right y-axis) for shape optimizations on mesh level L3 using step sizes, h=$10^{-1}$, $10^{-2}$, $10^{-3}$, $10^{-4}$, $10^{-5}$, and $10^{-6}$. The simulations are further desribed in Tab. 3 under the names: SPL3e1c, SPL3e2c, SPL3e3c, SPL3e4c, SPL3e5c, and SPL3e6c, respectively. As seen, the three optimizations with largest finite difference gradient step sizes converge deeply which is not true for the three optimizations with smallest step sizes.

**Table 6.** Overview of optimized shapes obtained from CFD-based shape optimizations of wind turbine blade tips. Operational conditions are found in Tab. 2. Naming convention explanation using **SPL1e3hb** as an example:

(**SP**:) a singlepoint optimization (**L1**:) on mesh level L1 (**e3**:) using a finite difference step size of $10^{-3}$ (**h**:) (hot-)started from the optimized shape on L2 (**b**:) with limited design variable bounds that work on all grid levels.

All optimizations were carried out using either 648 (L3) or 1296 (L2, L1) CPUs split into 12 groups - one group of either 54 (L3) or 108 (L2, L1) CPUs for each design variable.

| ID | Mesh level (See Tab. 3) | CPUs | Wall clock⋆ $10^{-7}$ threshold / full [h] | Design variables † $\begin{bmatrix} \theta_1, & \theta_2, & \theta_3, & \theta_4 \\ c_1, & c_2, & c_3, & c_4 \\ c_{wl}, h_{wl}, s_{wl}, R_{ext} \end{bmatrix}$ | $M_{\text{bending}}\vert_{0.9_{r/R}}$ (baseline fraction) | Mech.Power (improvement) |
|---|---|---|---|---|---|---|
| SPL3e3cb | L3 | 648 | 15.6 / 18.6 | $\begin{bmatrix} +4.0 & -10.0 & +5.0 & +5.0 \\ \cdot 1.0 & \cdot 1.0 & \cdot 1.0 & \cdot 1.0 \\ \cdot 0.1 & +0.0 & +0.1 & \cdot 1.00 \end{bmatrix}$ | 1.0000 | 0.01 % |
| SPL2e3cb | L2 | 1296 | 129.2 / 299.3 | $\begin{bmatrix} +4.7 & -10.0 & +5.0 & -4.5 \\ \cdot 1.0 & \cdot 1.0 & \cdot 1.0 & \cdot 0.8 \\ \cdot 0.0 & +2.0 & +2.0 & \cdot 1.1 \end{bmatrix}$ | 1.0000 | 0.42 % |
| SPL1e3cb | L1 | 1296 | 384.1 / 547.1 | $\begin{bmatrix} +4.1 & -10.0 & +5.0 & -4.0 \\ \cdot 1.0 & \cdot 1.0 & \cdot 1.0 & \cdot 0.7 \\ \cdot 0.0 & +2.0 & +1.9 & \cdot 1.1 \end{bmatrix}$ | 1.0000 | 0.44 % |
| | Evaluation of optimization result using a regenerated volume mesh: | | | | 0.9992 | 1.12 % |
| SPL1e3hb | L1 | 1296 | 204.3 / 332.4 | $\begin{bmatrix} +4.1 & -10.0 & +5.0 & -4.0 \\ \cdot 1.0 & \cdot 1.0 & \cdot 1.0 & \cdot 0.7 \\ \cdot 0.0 & +2.0 & +1.9 & \cdot 1.1 \end{bmatrix}$ | 1.0000 | 0.44 % |
| | Evaluation of optimization result using a regenerated volume mesh: | | | | 0.9992 | 1.12 % |

⋆ Given that some groups of CPUs on the HPC cluster will be faster than other CPU groups a representative computation speed ([it]/[sec]) for each grid level has been computed as an average over an entire optimization on a given grid level. This allows for a fair comparison between optimizations carried out on the same grid level although they have not been computed using the exact same CPUs.

† The lower and upper design variable bounds are: $\begin{bmatrix} -10.0, -10.0, -10.0, -10.0 \\ \cdot 0.5, & \cdot 0.5, & \cdot 0.5, & \cdot 0.5 \\ \cdot 0.0, & +0.0, & +0.0, & \cdot 1.0 \end{bmatrix} \leq \begin{bmatrix} \theta_1, & \theta_2, & \theta_3, & \theta_4 \\ c_1, & c_2, & c_3, & c_4 \\ c_{wl}, h_{wl}, s_{wl}, R_{ext} \end{bmatrix} \leq \begin{bmatrix} +5.0, & +5.0, & +5.0, & +5.0 \\ \cdot 1.0, & \cdot 1.0, & \cdot 1.0, & \cdot 1.0 \\ \cdot 1.0, & +2.0, & +2.0, & \cdot 2.0 \end{bmatrix}$





As seen from Tab. 6 a total of four optimizations where run: Three optimizations (SPL3e3cb, SPL2e3cb, and SPL1e3cb) were (cold-)started from a straight baseline blade whereas the final optimization (SPL1e3hb) used the optimization result from L2 as a starting point for an optimization on L1.

Starting from the left in Tab. 6 one can after **ID** and **Mesh Level** find two columns describing the computational resources used in this study (i.e., **CPUs** and **Wall clock**). When discussing the amount of CPUs used in the study it is well to remember that the CPUs are split into 12 groups; one group for each design variable to reduce the gradient computation time. The reason for using only $12 \cdot 54 = 648$ CPUs on mesh level L3 is that the efficiency study (see, Fig. 6) clearly showed that very little speed-up could be gained by increasing the amount of CPUs on this mesh level. Similarly, for L2 there is a drop in efficiency

in Fig. 6 after 108 CPUs which is why $12 \cdot 108 = 1296$ CPUs where used for SPL2e3cb. However, for L1 the efficiency in Fig. 6 for the EllipSys3D flow solver is at the upper possible limit and more CPUs could advantageously have been used. The reason for limiting SPL1e3cb and SPL1e3hb to 1296 CPUs has to do with the computational resources available for the present work. This also means that the computation time could be much improved: Simply by raising the number of CPUs from 108 to 432 one could gain a factor 4 in speed-up meaning that SPL1e3hb could be carried out in only $204.3/24/4 \approx 2$ days. For

a well converged high-fidelity optimization 2 days is certainly an acceptable computation time. One may find more optimistic computation time consumptions reported in the literature but typically with a correspondingly poor convergence of the design optimization problem. As also indicated in Fig.10 the computation time is very dependent on the chosen threshold. Therefore, one should not discuss one without considering the other.

  While on the topic of computation time it may be relevant to mention the $\tau$ time unit which is a non-dimensional work unit

to compare across HPC systems[13]. The Tau code was downloaded and 10 runs were carried out on the Sophia cluster resulting in an average execution time of: 3.89 seconds. Using this result one can now compute a unit less normalized version of the reported wall clock times in Tab. 6. As an example, the SPL2e3cb wall clock time to reach the threshold (129.2 hours) is in normalized $\tau$ units: $129.2 \cdot 60. \cdot 60.[\text{sec}]/3.89[\text{sec}] = 119.67 \cdot 10^3$.

  It is difficult to relate the reported timings to relevant literature from the wind energy community since very few high-fidelity

CFD-based shape optimization studies of that magnitude exist (Madsen, 2020, Tab. 3.1). Elfarra et al. (2014) do mention an approximate wall clock timing of 240 hours for their gradient-free optimization but since the timings are not in normalized $\tau$ units it is difficult to compare across HPC platforms. Furthermore, Elfarra et al. do not mention how well their optimization problem is converged in terms of optimality reduction which makes a comparison very difficult to carry out. Turning to the study by Dhert et al. one can, however, find both wall clock and optimality reduction reported: They spend 8.25 hours on a

wind turbine design problem with a $2.6 \cdot 10^6$ cell mesh resolution. The reported mesh resolution is most easily compared with the SPL2e3cb results in Tab. 6. However, it is difficult from the reported final optimality (Dhert et al., 2017, Tab. 2) to learn how many orders of magnitude it has been converged making it difficult to compare with the presented timings in this study. Madsen et al. (2019) report CPU timings (Madsen et al., 2019, Tab. 6) for adjoint-based high-fidelity shape optimizations using a single design variable of close to 60 hours for a mesh with resolution equal to L1 in Tab. 3. Again, these computations were

carried out on a different HPC platform than the present study so these timings are difficult to compare but as optimizations

---

[13]http://www.ipacs-benchmark.org/index.php?s=download&unterseite=bench, accessed Sep 30, 2021





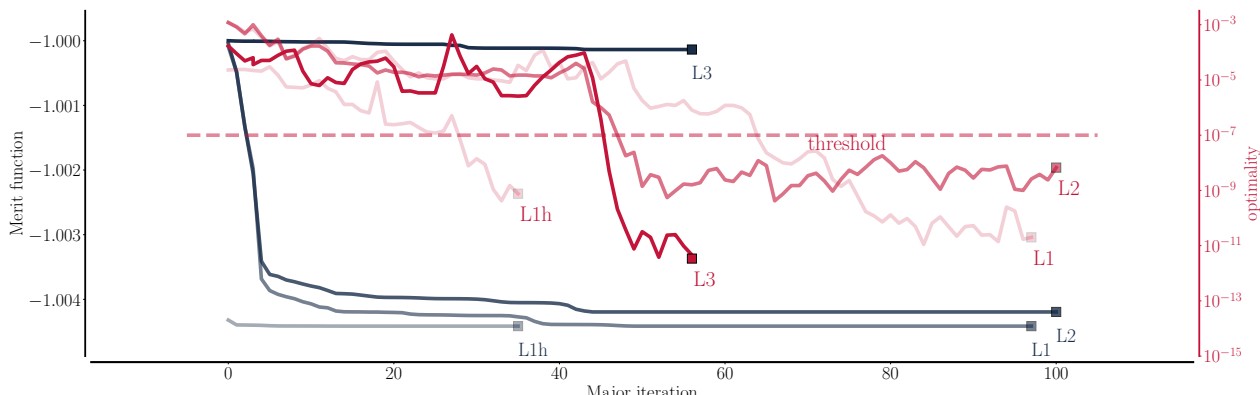

**Figure 10.** Merit function (black, left y-axis) and Optimality (red, right y-axis) for shape optimizations (Tab. 6) on mesh levels L3, L2, and L1 using finite difference step size, h=$10^{-3}$. The SPL1hb is a hot-started optimization on L1 where the result from L2 is used as starting point. All other optimizations start from a straight blade.

tend to become more difficult to converge as more design variables are included the 60 hours should certainly be seen as a lower possible bound in that study. In summary, a realistic lower bound for a high-fidelity optimization that is well converged seems to be around 2-3 days for very efficient frameworks.

Returning to Tab. 6 one can in the fifth column find the final design variables for each optimization. For readability all
design variables ending at the upper/lower limit have been colored accordingly. Overall, it can be said that the final shape trends favoured by the optimizer are an increase in sweep and an even greater increase in flapwise displacement. Both design variables are used to mitigate an increase in bending moment as the blade is extended. Of particular interest is the change in final sweep design variable from L2 to L1 where the L1 result, $s_{wl} = 1.9$, is not on the limit anymore and the optimizer seems to have found a maximum for this design variable. This aligns very well with findings in the related surrogate-based
study where a maximum for the sweep design variable is found at 2% d/R (Zahle et al., 2018, Fig. 4). Turning to the twist and chord design variables the optimizer favours the twist design variable over the chord design variable to shed loads. For all optimizations the optimizer favours curvature and for most optimizations it is at the very limit of what is allowed for the parametrization. All design trends agree with the surrogate-based design study by Zahle et al. (2018).

The resulting shape is visualized in Fig. 11 and an analysis of planform and spanwise forces is given in the following
subsection (Sec. 6.2.1). As seen, the produced novel curved tip shape is a means to extend the blade in an efficient manner and it effectively mitigates the downwash by displacing the tip vortex. The vortex is somewhat diffused and smeared out along the extended tip structure protruding into the out-of-plane region as one would expect to see for a functioning optimization.

With respect to the final two columns in Tab. 6 showing bending moment constraint and gained improvement they agree well with expectations: The bending moment constraint is indeed a design-driving load constraint and will to a great extend dictate
how far the blade can be extended. In practice, the load level constraint would depend on the structural capacity in the blade, which may well allow for increases in the flapwise moment. A relevant investigation would therefore be to explore the effect of





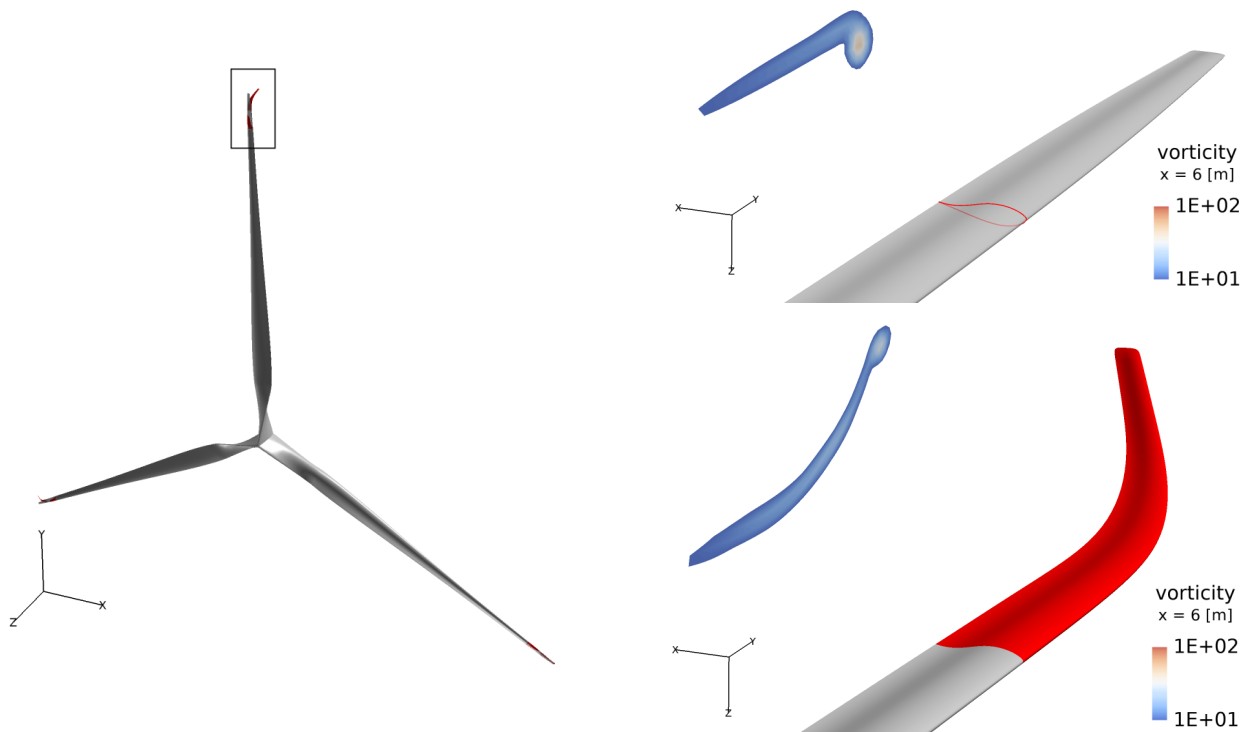

**Figure 11.** Tip vortex visualization using the shape optimization result from grid level L1. Wind direction is along the $z$-axis. The full rotor image (left) is a superimposed image showing both the baseline rotor (gray) with straight blades and the final rotor shape (red) with the optimized tip shapes. The tip vortex is visualized both for the straight baseline (right, top) and for the final optimized tip shape (right, bottom) to show that it is smeared out and moved away from the rotor plane.

varying this constraint, which however, is beyond the scope of this work. To give the optimizer more freedom to operate future studies will involve relaxing this constraint to explore the resulting shapes produced by the optimizer.

With respect to the gained improvement in mechanical power it can be said that the optimization results from the finer grid
levels (SPL2e3cb, SPL1e3cb, and SPL1e3hb) agree very well on $0.42 - 0.44\%$ improvement where the result from L3 is much lower. This observation agrees with earlier findings (Madsen et al., 2019) stating that very coarse mesh levels should be used with care in shape optimizations. Furthermore, it should be pointed out that the SPL3e3c improvement percentage from Tab. 5 is much higher than the SPL3e3cb improvement percentage from Tab. 6 because the design variable bounds for $h_{wl}$ and $s_{wl}$ were changed to 3.5 from 2.0. Thus, one cannot directly compare SPL3e3c and SPL3e3cb.
Finally, while on the topic of improvement it is well to note the importance of regenerating the mesh: As seen, the final shape design from SPL1e3cb and SPL1e3hb are for completeness evaluated with a regenerated volume mesh to ensure as accurate a result as possible. As a result, the final improvement in mechanical power changes from $0.44\%$ to $1.12\%$. Importantly, the





bending moment fraction constraint only changes from 1.000 to 0.9992 signifying that the final shape is still feasible and at the upper limit of the constraint as expected.

The 1.12% improvement in mechanical power aligns very well with previous findings from the literature (see, e.g., Tab. 1): Matheswaran et al. (2019) report a 2.5% power increase for a load neutral optimization and Zahle et al. (2018) report an improvement in power of 0.76% for straight blade extensions and up to 2.6% improvement for optimized winglet shapes. The surrogate-based approach by Zahle et al. (2018) used the exact same mesh generator and flow solver meaning that the results should align fairly well. Given that the developed FFD-based parametrization in the present study does not produce true $90°$

winglet shapes (see, Fig. 15 (left) for a comparison) it is reasonable that the expected improvement from the present study should lie somewhere between 0.76% and 2.6% which is also the case.

    The reason that mechanical power changes much more than the bending moment when the mesh is regenerated is that the bending moment computation is driven by pressure and friction whereas the mechanical power computation is based on drag which is much more mesh quality dependent. The same phenomenon is seen in standard 2-D CFD airfoil computations where

lift mainly is a projection of pressure forces whereas the viscous forces are very important for the drag. A small change in force vector will therefore mean much more for the drag than for the final lift.

    It should be clearly stated, that the change in final mechanical power in Tab. 6 from 0.44% to 0.12% deserves further investigation in future studies. Below, some of the planned investigations are described in further detail.

    One could opt to restart the optimization on a regenerated mesh around the final L1 shape to obtain a result that is more

independent of mesh quality. Alternatively, one could look into other popular methods such as radial basis functions or the inverse distance method to investigate whether these methods produce meshes that are closer in quality to an actual regenerated mesh. Yet another option is to regenerate the mesh after every optimization step. However, given that the aim is to arrive at a high-fidelity optimization framework that also utilizes adjoint solvers that is not a desirable avenue to pursue since it would further complicate the gradient computation. A final option worth mentioning is to make a dedicated L3 mesh that is not a

result of the grid coarsening described in Sec. 4.3. A better way to generate the L3 mesh would be to grow the mesh directly on L3 using the hyperbolic mesh generator. This approach has proved to be very efficient in the past. Once this improved L3 mesh is taken into use one can carry out the start of the optimization on L3 before proceeding to L2 and in turn L1 thus saving further time. For the present setup the best approach seems to be starting an optimization on L2 (SPL2e3cb) and then proceeding on L1 (SPL1e3hb) using the L2 result as seen in Fig. 12. Importantly, the final shape is the same when starting from a straight

blade (SPL1e3cb) or from an L2 result (SPL1e3hb) as one would expect for a well-functioning setup. Given that a speed-up of about a factor 2 is observed this grid sequencing approach in shape optimization seems very advantageous.

### 6.2.1   Analysis of the optimized shape

To inspect the optimized blade shape the planform is visualized in Fig. 13 where after the resulting spanwise forces are visualized in Fig. 14.

Inspecting the final twist distribution (Fig. 13, upper) a slightly jagged curve is seen where the optimizer tries to continue the upward rising twist curve at the start of the tip as well as tries to introduce more negative twist towards the very tip of the



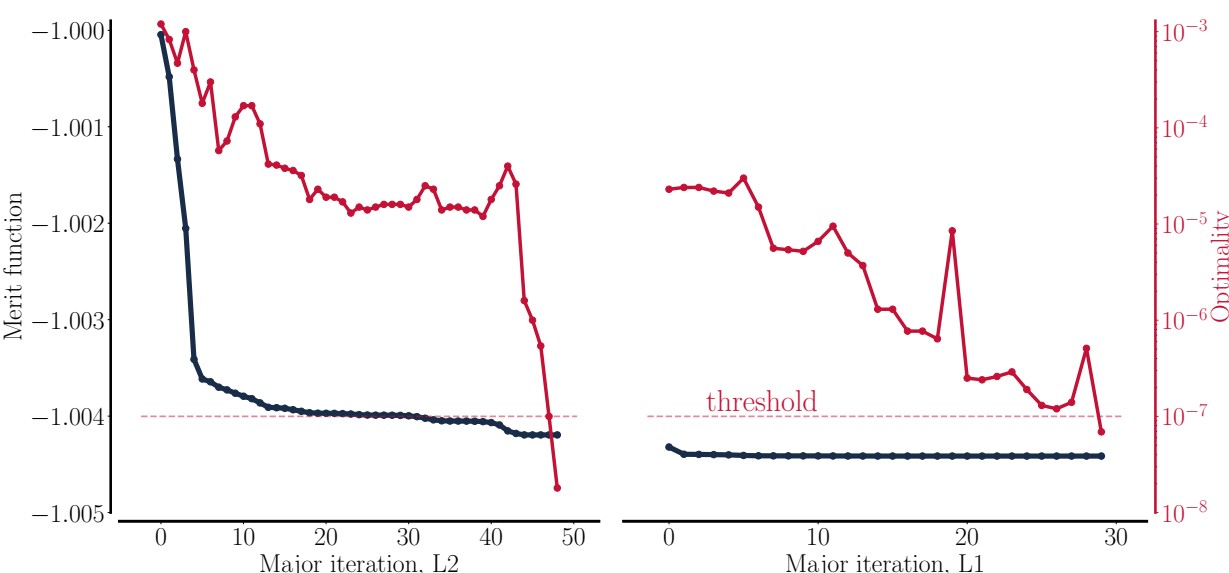

**Figure 12.** Full optimization procedure using grid sequencing: The result from grid level L2 (left) is used as a starting point on grid level L1 (right) to save time. The optimizations are called SPL2e3cb and SPL1e3hb and also figure both in Tab. 6 and in Fig. 10.

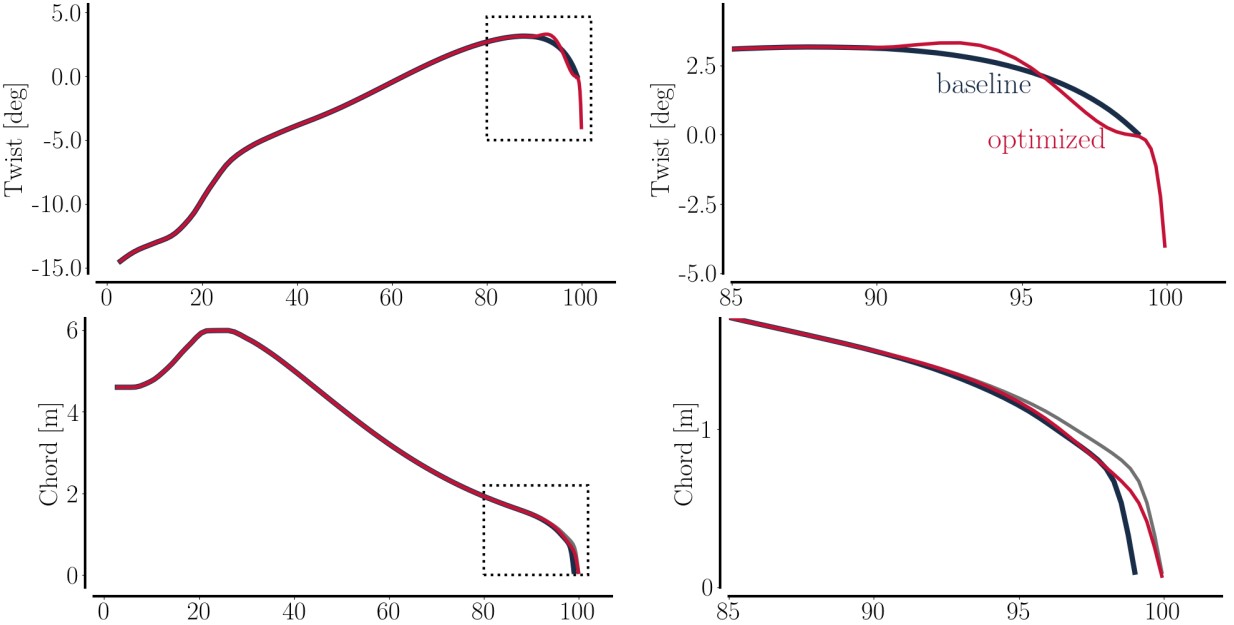

**Figure 13.** Planform distribution for twist and chord for the shape optimization result from grid level L1. Gray chord distribution signifies a stretched blade without any chord scaling in order to better discern where the reduced chord is taking place on the stretched blade.





blade. Reassuringly, these exact same trends were observed in a recent surrogate-based study (Zahle et al., 2018, Fig. 8, right) albeit with a more smooth curve. The present study's parametrization has mainly been designed with robustness and mesh quality in mind focusing on avoiding negative cells on the finest mesh level. Future work will entail experimenting with the

parametrization in the attempt to try to arrive at a more smooth twist curve. That being said, the resulting spanwise forces are very smooth even for the present setup as can be seen in Fig. 14.

The chord distribution only changes at the very tip of the blade (Fig. 13, lower) where the outermost chord design variable is used to slim the blade. One could attempt to move the second outermost FFD-box section towards the tip to activate the remaining chord design variables. This would indeed also give the optimizer more freedom to shape the blade tip. However,

that FFD-section has been purposefully placed at a distance to the actual blade tip to protect the cell volumes at the very tip meaning one should use caution not to compromise the mesh quality. Overall, the trend of slimming the blade using the chord design variable as the blade is extended is to be expected for these optimizations.

Finally, the resulting spanwise forces are shown in Fig. 14 where the driving force is placed above the normal force. The entire span is shown to the left and a zoom of the outermost part of the blade is shown to the right.

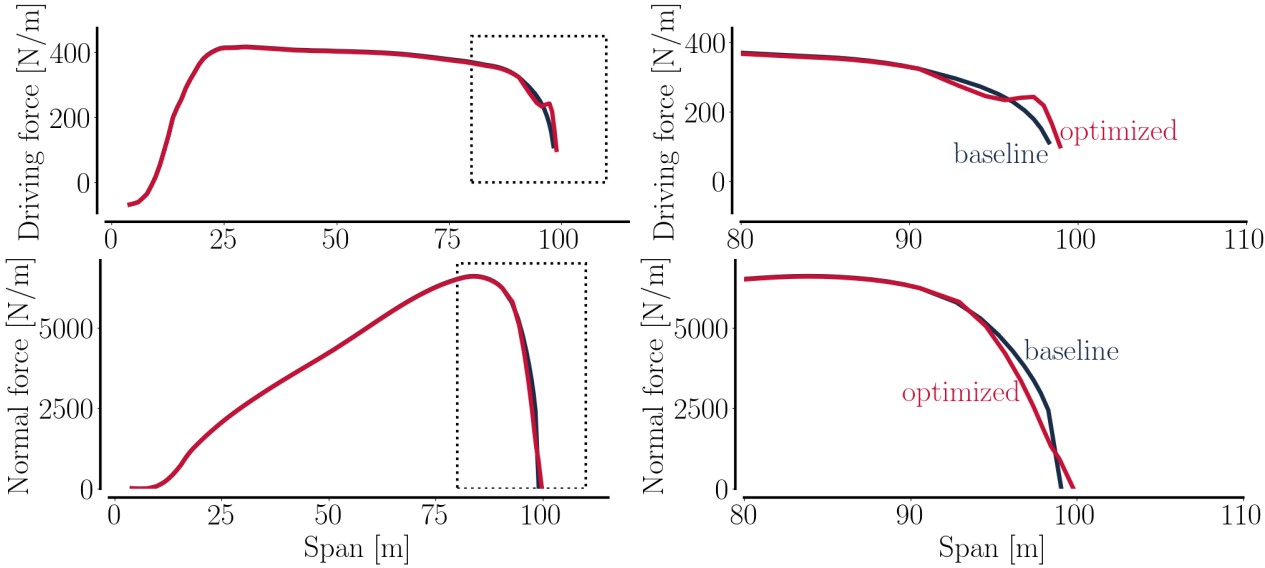

**Figure 14.** Driving- (top) and normal (bottom) forces for the shape optimization result from grid level L1. Black dashed rectangles (left) indicate limits for the zoom plots (right).

The driving force is exactly as expected for this type of tip optimization: As visible in the zoom plot (right) the optimizer sacrifices a small portion of power at the start of the tip (90-95 [m]) but generates more power towards the very tip of the extended blade, resulting in a net increase in torque.

Turning to the normal force for the optimized shape the tip is effectively de-loaded mid tip ($\sim 97$[m] span) after an initial slight increase at the beginning of the tip. The de-loading allows the optimizer to extend the blade thus incurring new loads at

the very tip.





In sum, a novel curved tip shape has been designed. The results show that with approximately 1% blade extension, 2% flapwise displacement, and slightly below 2% edgewise displacement one can obtain a 1.2% increase in power. The design favours as much curvature as is possible with the present parameterization and it is likely that parameterizations allowing for a curvature closer to a 90° winglet-like shape would find an even greater power increase. Indeed, the novel shape is able to

extend the blade efficiently in the following manner: Using a combination of the twist and chord design variables, it effectively sheds loads at the beginning of the tip (Fig. 14) where also a reduced driving force is observed in the process, while extending the blade with a slender chord distribution, minimising the load impact from the extension. Flow visualizations (Fig. 11) showed that compared to the original tip, the curved tip shape results in a more smeared out tip vortex, thus reducing tip loss. However, further investigations are needed to fully understand how for example the addition of sweep at the tip is favourable

to a non-swept tip from an aerodynamic point of view.

## 6.3 A comparison across fidelities

Given that the study by Zahle et al. (2018) is closely related to the present work it is rewarding to compare the parametrizations and in particular the difference in maximally allowed curvature by superimposing a tip shape result from the present work (red) on to the final design (gray) from the surrogate-based study by Zahle et al. as seen in Fig. 15. Inspecting the left plot in Fig. 15

it is evident that the present study cannot produce true 90° winglet shapes due to concerns of the mesh quality which is why one should expect to find final improvements slightly lower than in the study by Zahle et al. Reassuringly, a 1.2% increase in power is reported in the present study whereas a 2.6% increase in power is reported in Zahle et al. (2018).

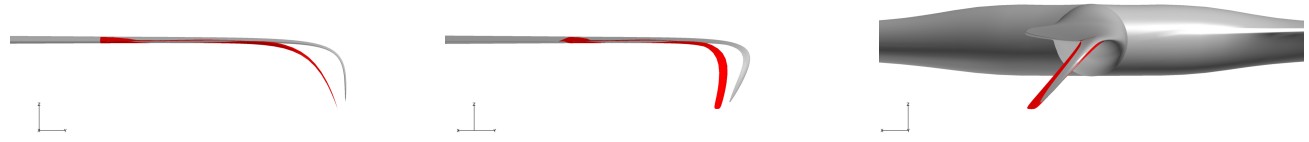

**Figure 15.** Resulting tip shape from the present study (red) compared to the surrogate-based design by Zahle et al. (gray). Both results exhibit maximally allowed curvature by their respective parametrization. As seen, the curvature from the present work (red) is not a true 90° winglet as in the study by Zahle et al. (2018) signifying that one should expect a difference in final possible improvement (final results are 1.2% and 2.6%, respectively).

## 6.4 Future work

As a first step one should further increase the robustness and efficiency of the presented FlowOpt framework. This includes

adding enhanced convergence methods and an adjoint method.





An enhanced convergence method would apart from a general increase in optimization robustness also ensure that aerodynamically challenging shapes due to, e.g., stall, could be handled (See, e.g., Fig. 16) meaning that a significant increase in the design space could be gained as it would only be due to negative cells that one would have to limit the design variables.

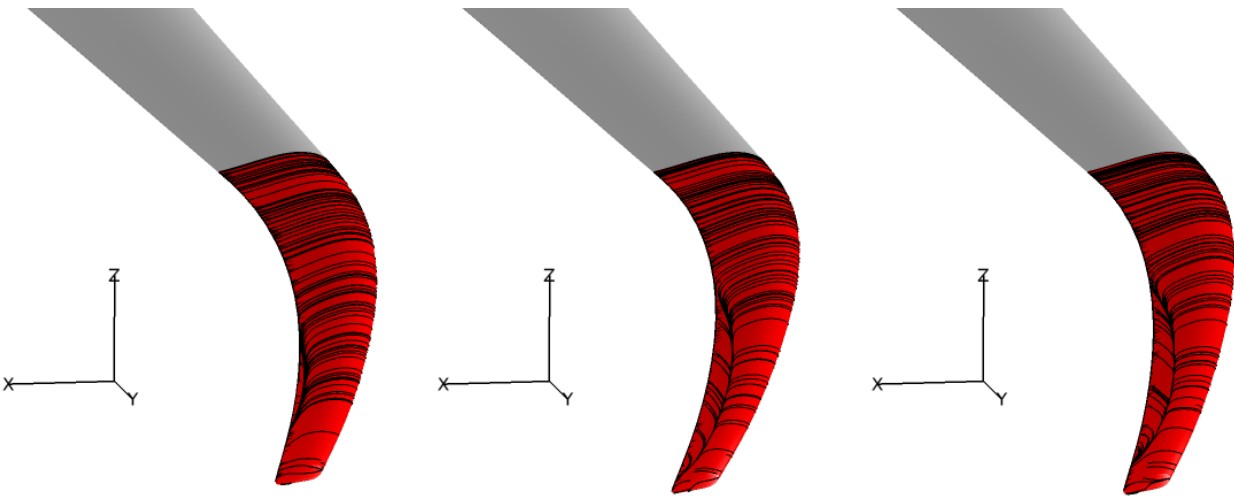

**Figure 16.** A sequence of three different deforming curved tip shapes (red) imposed on the baseline blade (gray) where surface restricted streamlines can be seen as thin black lines. The flow visualization shows an emerging stall region leading to the impaired flow convergence. Shedding of vorticity is introduced as the blade is further twisted. These shapes were created by the optimizer during an early exploratory optimization.

The adjoint method allows for a gradient computation time that is independent of the number of design variables which would be highly relevant for future work. Notice, that the adjoint method will most likely not result in a speed-up of what can already be achieved for the presented optimizations with the current framework when using the parallelization techniques in OpenMDAO to compute perturbations for various design variables simultaneously. In fact, given that not all adjoint solvers are as fast as their flow solver counterparts on might see a slight slow-down. However, given that the present study uses the finite difference method which clearly depends on a preceding step size study the final runtimes are difficult to predict and discuss without a concrete comparison being done.

Leaving the discussion of possible lower bounds for runtimes aside, one thing that is certain is that the framework with an adjoint solver will be able to take on new optimization problems altogether since the gradient computation will be independent of the number of design variables. Thus, full shape optimizations using free-form techniques will be manageable - something one can never hope to do with the finite difference method.

## 6.5 Learning outcomes

Based on the above-described detailed high-fidelity shape optimization study these are the overall findings:



- A thorough literature review showed that there is a lack of high-fidelity shape optimization studies within wind energy where most works simply are parameter studies.

- Robust mesh deformation is an absolute key feature for high-fidelity shape optimization with $y+ \sim 1$ and $\mathcal{O}(10^7)$ cells.

- In order to explore a larger design space (e.g., stall regions) there is a need for enhanced convergence methods which would also bring about an increase in robustness.

- Meticulous setup of the finite difference method will allow for deeply converged design optimization problems even without machine accurate gradients.

- Although the finite difference method is a viable approach high-fidelity shape design, the authors can conclude based
on experience with both direct CFD-based optimizations and surrogate-based optimizations that due to ease of use and a much lower computational cost one should prefer the surrogate-based approach for optimizations up to about a dozen design variables if one can accept the drop in model fidelity.

## 7 Conclusions

In this study a novel curved tip shape was aerodynamically designed for maximum power using a CFD solver on a 10 MW
reference wind turbine with the constraint that the initial steady-state loads should not be compromised. The study showed that a $1.2\%$ increase in power was possible while satisfying the imposed constraints on loads and geometry. The final curved tip results in a $1\%$ blade length extension, a $2\%$ flapwise tip displacement, just below $2\%$ in edgewise tip displacement, and as much tip curvature as possible within the developed parameterization. Using twist and chord design variables to reduce loads and slim the outermost part of the blade, the novel curved tip shape efficiently extends the blade and a flow analysis visualized
how the final design effectively displaces the tip vortex to mitigate induced drag. A tip design as the one presented, could be mounted on already installed wind turbines as a sleeve-like solution, or be conceived as part of a modular blade with tips designed for site-specific conditions. Importantly, this study was not aeroelastic but aerodynamic only. Only steady-state flow conditions and normal operation were considered, and a detailed unsteady load analysis based on the IEC standard was thus not carried out, which would be needed to design the structural geometry of the blade tip, and ensure that it is indeed load
neutral.

Turning to the numerical aspect of the study it can be concluded that it is indeed possible to tightly converge direct CFD-based design optimization problems using the finite difference method as long as a meticulous step size study is carried out. However, the finite difference method is found to be extremely expensive on industrial scale cases (above $14$ million cell meshes) and a surrogate-based approach should be favoured due to ease of use and implementation as long as a drop in
model fidelity can be accepted. Furthermore, the study revealed that robust mesh deformation routines are very important for a successful optimization framework.

A comprehensive literature review on blade tips preceding the optimizations revealed many overall favourable design trends could be identified. Furthermore, it was found that up to $2.5 - 2.6\%$ increase in power should be possible for winglet-like load



neutral tip shapes. However, as also pointed out in the literature review there is a void of high-fidelity shape optimization results
and this study shows at least one way to setup the essential components in a framework for CFD-based optimization and how
to tightly converged the design optimization problems through a meticulous fine tuning of the setup.

*Data availability.*   Data is available upon request to corresponding author.

## Appendix A: Visualization in the rotor plane of the optimized tip shape

*Author contributions.*   MHAM carried out the literature review, wrote the FFDlib, ran all optimizations and wrote the bulk of the paper. FZ
developed PGL and assisted in work related to the parameterization and optimization. Also, FZ assissted in postprocessing and analysis of all
results and in the comparison to surrogate-based optimizations. SGH provided meshes and helped guide the study - particularly in the early
stage (develop parameterization, ensuring cross-platform compatibility, etc.) and assisted in analysis of results. Furtermore, SGH ensured the
final parameterization is aligned with the approach used in the DTU Wind Energy FSI framework for future work. TKB guided the project,
assisted with expert knowledge on blade tip design and aligned the present study with the surrounding SmartTip study. NNS as one of the two
original EllipSys3D developers, advised on computational setup, parallelization and implemented the enhanced mesh deformation algorithm
used in the present study. He also worked on the EllipSys3D coarse grid solver complexification. All authors took part in writing and editing
the paper.

*Competing interests.*   The authors declare that they have no conflict of interest.

*Acknowledgements.*   This study was funded through the SmartTip project[14]. The SmartTip project involved one industrial partner[15] and was
funded by Innovation Fund Denmark[16] (contract number 7046-00023B).
A special mentioning of Head of Section (ARD) Flemming Rasmussen, DTU Wind Energy, is also warranted. He has through several
years supported the development of a high-fidelity design framework without which, this study would not have been possible.
Finally, the authors thank PhD student Neil Wu (MDOLab, University of Michigan) and OpenMDAO Project Lead Justin S. Gray (NASA,
Glenn) for assistance with pyOptSparse and OpenMDAO, respectively.

---

[14]https://www.vindenergi.dtu.dk/english/research/research-projects/smart-tip, accessed Sep 30, 2021
[15]https://www.siemensgamesa.com/en-int, accessed Sep 30, 2021
[16]https://innovationsfonden.dk/en, accessed Sep 30, 2021





**Figure A1.** Baseline (gray) and optimized (red) blade shape from different angles.





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
