# Peer review of "CFD-based curved tip shape design for wind turbine blades"

_Wind Energy Science, 2021_

## Referee Comment (RC2)

The authors presented a very interesting and valuable study on optimization of WT blade tip. Particularly, it seems that the main contribution of the research is on proposing a novel methodology which can be implemented in the process of optimization. Through employing that process, 12 design variables which is a considerable number of variables in comparison with previous investigations on blade tip shape, have been taken into account. Their results show that for the optimized geometry of the blade tip, the power output of the turbine has been increases 1.2% while there is no excessive bending moment at the tip location, i.e. top 10% of the blade length. The manuscript is very well written and structured. The authors have provided a very comprehensive literature review with regard to the corresponding research area. However, there are few issues need to be resolved and clarified to improve the article.  The comments are summarized as follows:

- Since the literature review is extensive and it occupies a big portion of the manuscript, the reader might be confused about the novelty of the paper at the end of "literature review" section. I would suggest to re-state the novelty and contribution of the study at the end of this section.
- At page 18, line 535, it is mentioned that the flow over blade has been considered to be fully turbulent (which is true!). It would be more informative to include the physical justification behind that assumption.
- It is indicated in the manuscript that steady-state flow modeling has adopted to solve the equations. However, as you confirm, tip vortices are unsteady phenomenon in nature and thus steady simulations might affect significantly the results. How do you justify this issue? Is there any other previous investigation that clearly addresses that effect?
- In figure 3, it would beneficial for the reader to see the boundary conditions in the figure where the domain is displayed.
- The operating conditions of the given turbine has not been clearly presented. For instance, it would be great to include the power curve of the turbine (Cp ~ TSR), so the reader can identify that the rotational speed at which your simulations are performed, is lower than the optimum TSR or higher. TSR as a governing parameter of the fluid flow around the turbine, significantly influence the flow structures at the blade location since it determines the angle of attacks experienced by the blade at different sections.
- Finally, although the focus of this study is on the methodology and its effectiveness, it would be crucial to validate the numerical results against any available data. Particularly, because the authors concluded about the correctness of the results obtained from the optimization process, i.e. 1.2% increase in power output. Since the simulations are not performed in unsteady-state mode and the results have not also been validated, the

increase in the power obtained from the optimization process might not be reliable.

---

## Author Response (AR1)

Review response for

**CFD-based curved tip shape design for wind turbine blades**

Mads H. Aa., Madsen[1], Frederik, Zahle[1], Sergio G., Horcas[1],
Thanasis K., Barlas[2], and Niels N., Sørensen[1]

[1]Aero- and Fluid Dynamics (AFD) section, DTU Wind Energy,
Lyngby  Campus, Nils Koppels Allé, building 403, 2800 Lyngby,
Denmark
[2]Airfoil and Rotor Design (ARD) section, DTU Wind Energy,
Risø Campus, Frederiksborgvej 399, 4000 Roskilde, Denmark

March 2022

All comments will be adressed using the following format:

Comment XX:
Response:
Action:

**Reviewer #1 comments and response**

Comment 01:
"This manuscript presents an optimization framework for tip designs in wind turbines using CFD. The authors are attempting to come up with a load neutral tip shape for wind turbines using a direct high-fidelity CFD based approach. Only aerodynamic effects are considered. The authors begin with an extensive literature review that places the current work in context with archival literature. They divide recently published literature into parametric studies and optimization studies. They point to the need for higher model fidelity as cited by several previous studies, as well as the need for optimizations with unifying design optimization problems. These are the drivers for this study. The optimization method involves a straightforward process of determining design variables, calculating a deformed surface mesh, subsequently calculating a flowfield, functions of interest and constraints using EllipSys3D. This cycle is repeated until an optimized shape is obtained. The IEA 10MW is used as the baseline, and the design

optimization problem is performed on this turbine. The results are presented in terms of optimal step size, and finally the optimized tip shape compared to the baseline. The authors conclude by pointing out that mesh deformation and setting up the finite difference method carefully are critical in a CFD based approach, but ease of use may mean surrogate based approaches may be more viable currently."

Response:

The reviewer succinctly summarized the paper. We thank the reviewer for all the presented comments below and find they improve the final manuscript considerably.

Action:

N/A

Comment 02:

"The complex step method is used to compute reference gradients. However, it is also mentioned that there is a lack of robustness in the authors' implementation of the complex step method. This statement will need to be clarified to build confidence on the accuracy of reference gradient and the results presented here."

Response:

The Complex-Step implementation in EllipSys3D has already been verified with adjoint gradients showing up to 16 digits accuracy and is indeed fully machine accurate [3, Fig. 7.9]. The adjoint implementation in EllipSys3D is currently being prepared for presentation in another publication and it is outside the scope of the present publication to also include that. Below follows an explanation for the observed issue with the current Complex-Step implementation in EllipSys3D:

Throughout the course of an optimization the flow solver encounters several hundred different volume meshes and a select few of these (less than 5) have proven difficult to converge for the complexified flow solver. This is typically seen for large deformations where volume cells are particularly skewed. Thus, the baseline mesh is not affected. These few meshes that are difficult to converge are only a fraction of all the meshes seen throughout an optimization but they will still result in an optimization crash. Therefore, shape optimizations using Complex-Step variables were left out of the scope of the current publication until all minor issues were fixed. The reviewer is, however, correct that the phrasing in the submitted manuscript was misleading and did not capture the above-described explanation.

Action:

The manuscript was rephrased in Sec. 3.1.

Comment 03:

"The variables in the design optimization process are not clearly explained. I assumed $\Theta_{1,2,3,4}$ and $c_{1,2,3,4}$ are twist and chord for four points along the span

in the tip region(outer 10%), but this is not stated clearly and locations not given."
Response:
The reviewer is correct that $\Theta_{1,2,3,4}$ and $c_{1,2,3,4}$ are twist and chord as seen in Tab. 4. The reviewer also is correct that this could be better explained.
Action:
Text has been updated in the beginning of Sec. 5.1 for clarification. As explained in the text, the design variables are placed on FFD box sections, $S_{3,4,5,9}$. Normalized section coordinates in range [0;1] showing the placement of all FFD box sections have been superimposed on Fig. 7 to accommodate the clarification.

Comment 04:
"It is not clear how c is scaled. Is it with respect to chord at 0.9r/R?"
Response:
The authors agree this should be further clarified. The twist and chord variables are imposed section-wise on section $S_3$ ($\theta_1$, $c_1$), $S_4$ ($\theta_2$, $c_2$), $S_5$ ($\theta_3$, $c_3$), and $S_9$ ($\theta_4$, $c_4$) in Fig. 7, respectively. Sections $S_{6-8}$ are given from interpolation once section $S_5$ and $S_9$ are set.
   As an example:
The starting value for all chord-scaling variables is unity. If $c_1$ is reduced from 1.0 to $c_1 = 0.5$ it means that the chord has been scaled to 50% of its original size at section $S_3$ where this design variable is enforced.
Action:
Clarification has been inserted Sec. 5.1.

Comment 05:
"Recommend using the term merit function when discussing design optimization problem for consistency, rather than just in the results."
Response:
The authors only partially agree: It is correct that the distinction between merit function and objective function was never properly explained. This led to confusion and should be clarified. However, Sec. 5 is on the mathematical problem, and when presenting that the correct term is indeed 'objective function'. The term 'merit function' is only used in Sec. 6 when presenting the results since it is this term that the SNOPT optimizer exports. In order to properly be able to reproduce the results it is therefore 'merit function' that is the correct term in Sec. 6.
Action:
Requested clarification have been inserted at the end of Sec. 6.1 in the manuscript (see below):
"In order to accommodate reproducibility, Fig. 9 shows a 'merit function' instead of the actual design optimization problem objective since it is the former metric that SNOPT exports. However, these two metrics are highly related as detailed in SNOPT's manual and the merit function will converge to the objective function value as the solution is approached."

Comment 06:
"An iso-view of the blade in comparison to Zahle (2018) may be more informative than current tile 3 of figure 15."
Response:
The authors also find an iso-view could be interesting.
Action:
An new tile has been inserted in Fig. 15.

Comment 07:
"While this manuscript has been written well for most part, there are areas where typing errors have crept in or word choices cause confusion. A few examples: line 75 (also), line 241 (I'm not sure unimodal is the right word here), line 309, line 348, line 479, line 629, line 819 and so on. Recommend careful proofreading once again and rewording the unclear sentences."
Response:
The authors thank the reviewer for the meticulous review.
Action:
Edits have been inserted. Another careful proofreading was done as well (See section 'Other changes to the manuscript' further down).

Comment 08:
"In general, this is a well-written and researched paper and will advance the literature on CFD based design approaches for blade tip devices. The current study can be considered a companion study to the ones by Zahle et al. (2018) and Barlas et al. (2021) where the authors also tackle the problem of load neutral blade tip extensions. The paper addresses questions relevant to the scope of WES and presents a framework for using a CFD based design optimization process. I recommend that with minor revisions (addressing the concerns listed above), the paper be accepted for publication in WES."
Response:
Again, the authors thank the reviewer for the edits and minor revision resulting in an improved manuscript.

**Reviewer #2 comments and response**

Comment 01:
"The authors presented a very interesting and valuable study on optimization of WT blade tip. Particularly, it seems that the main contribution of the research is on proposing a novel methodology which can be implemented in the process

of optimization. Through employing that process, 12 design variables which is a considerable number of variables in comparison with previous investigations on blade tip shape, have been taken into account. Their results show that for the optimized geometry of the blade tip, the power output of the turbine has been increases 1.2% while there is no excessive bending moment at the tip location, i.e. top 10% of the blade length. The manuscript is very well written and structured. The authors have provided a very comprehensive literature review with regard to the corresponding research area. However, there are few issues need to be resolved and clarified to improve the article. The comments are summarized as follows:"

Response:

The authors thank the reviewer for a thorough review and find the resulting manuscript much improved.

Action:

N/A

Comment 02:

"Since the literature review is extensive and it occupies a big portion of the manuscript, the reader might be confused about the novelty of the paper at the end of "literature review" section. I would suggest to re-state the novelty and contribution of the study at the end of this section."

Response:

Agreed.

Action:

Inserted. See the very end of Sec. 2.

Comment 03:

"At page 18, line 535, it is mentioned that the flow over blade has been considered to be fully turbulent (which is true!). It would be more informative to include the physical justification behind that assumption."

Response:

In this work, the wall resolved boundary layers are assumed to be fully turbulent (modelled with the K-omega SST model), and as such laminar-to-turbulent transition is not taken into account. Depending on the surface conditions and type of airfoil, this assumption is not correct, since laminar-to-turbulent transition will not take place at the leading edge, but at some point along the chord. However, one can consider this modeling choice a conservative choice to ensure robustness of the design under conditions where the boundary layers are turbulent, for example due to surface soiling or erosion. It should also be added that including laminar-to-turbulent transition in a gradient based optimization context is quite challenging due to the inherent physics of transition that can lead to lack of numerical convergence to a steady state solution, due to unstable transition points, which we have discussed in the paper to be important for convergence of the numerical optimization. As such we are at present not able

to include transition in the shape optimization framework.
Action:
The physical justification has been inserted in Sec. 3.3.

Comment 04:
"It is indicated in the manuscript that steady-state flow modeling has adopted to solve the equations. However, as you confirm, tip vortices are unsteady phenomenon in nature and thus steady simulations might affect significantly the results. How do you justify this issue? Is there any other previous investigation that clearly addresses effect?"
Response:
It is correct that unsteady simulations would be necessary to accurately study, e.g., vortex breakdown in the wake further down stream. However, this paper is about optimization of the steady-state performance of the rotor. For this reason, the authors also limited the design space to avoid going into deep stall (visualized in Fig. 16) where the chosen steady-state model would be more challenged.
Action:
The consideration was included in the caption for Fig. 16.

Comment 05:
"In figure 3, it would beneficial for the reader to see the boundary conditions in the figure where the domain is displayed."
Response:
Agreed.
Action:
Clarification inserted in Fig. 3 caption.

Comment 06:
"The operating conditions of the given turbine has not been clearly presented. For instance, it would be great to include the power curve of the turbine (Cp TSR), so the reader can identify that the rotational speed at which your simulations are performed,is lower than the optimum TSR or higher. TSR as a governing parameter of the fluid flow around the turbine, significantly influence the flow structures at the blade location since it determines the angle of attacks experienced by the blade at different sections."
Response:
The authors agree that TSR is an important parameter. The rotor is operated according to the normal operating conditions. The requested power curve has been reported elsewhere [2, Fig. 33].
Action:
Tab. 2 listing operational conditions has been extended with a column for TSR.

Comment 07:

"Finally, although the focus of this study is on the methodology and its effectiveness, it would be crucial to validate the numerical results against any available data. Particularly, because the authors concluded about the correctness of the results obtained from the optimization process, i.e. 1.2% increase in power output. Since the simulations are not performed in unsteady-state mode and the results have not also been validated, the increase in the power obtained from the optimization process might not be reliable."

Response:

A comprehensive experimental validation is outside the scope of this paper which is a numerical study. That being said, the SmartTip project did include many other studies: A particularly relevant study was the wind tunnel testing of swept tip shapes [1] which the authors agree should have been cited.

Action:

The experimental study has been referenced at the end of Sec. 6.4.

**Other changes to the manuscript**

Author comment 01:

There was a typo in the conclusion mentioning 1.2% instead for 1.12%.

Action:

Fixed

Author comment 02:

Results for the SPL3e3cb optimization were re-run since it was found out that an outdated was used mesh. After the re-run the L3 results match much better with the remaining results in Tab. 6 as one would also expect based on the presented mesh convergence study (Tab. 3). It also made it relevant to carry out a hotstart from L2 (see SPL2e3hb in Tab. 6). Reassuringly, the final shape results on L2 are identical with or without hotstarting as one would expect. This was also shown for L1 in the original manuscript.

Action:

Updated Tab. 6 and Fig. 10 accordingly. Accompanying text was also updated.

**References**

[1] T. Barlas, G. R. Pirrung, N. Ramos-García, S. G. Horcas, R. F. Mikkelsen, A. S. Olsen, and M. Gaunaa. Wind tunnel testing of a swept tip shape

and comparison with multi-fidelity aerodynamic simulations. *Wind Energy Science*, 6(5):1311–1324, 2021.

[2] Pietro Bortolotti, Helena Canet Tarrés, Katherine Dykes, Karl Merz, Latha Sethuraman, David Verelst, and Frederik Zahle. Iea wind tcp task 37: Systems engineering in wind energy-wp2.1 reference wind turbines. Technical report, 2019.

[3] Mads Holst Aagaard Madsen. *High-Fidelity CFD-based Shape Optimization of Wind Turbine Blades*. PhD thesis, Denmark, 2020.